# LEARNING INSTANCE-SOLUTION OPERATOR FOR OPTIMAL CONTROL

## ABSTRACT

Optimal control problems (OCPs) involves finding a control function for a dynamical system such that a cost functional is optimized, which are central to physical system research in both academia and industry. In this paper, we propose a novel instance-solution operator learning perspective, which solves OCPs in a one-shot manner with no dependence on the explicit expression of dynamics or iterative optimization processes. The design is in principle endowed with substantial speedup in running time, and the model reusability is guaranteed by high-quality in- and out-of-distribution generalization. We theoretically validate the perspective by presenting the approximation bounds for the instance-solution operator learning. Experiments on 7 synthetic environments and a real-world dataset verify the effectiveness and efficiency of our approach. The source code will be made publicly available.

## 1 INTRODUCTION

The explosion of data for embedding the physical world is reshaping the ways we understand, model, and control dynamical systems. Though control theory has been classically rooted in a model-based design and solving paradigm, the demands of model reusability, and the opacity of complex dynamical systems call for a rapprochement of modern control theory, machine learning, and optimization. Recent years have witnessed emerging trends of control theories with successful applications to engineering and scientific research, such as robotics (Krimsky & Collins, 2020), aerospace technology (He et al., 2019), and economics and management (Lapin et al., 2019) etc.

We consider the well-established formulation of optimal control (Kirk, 2004) in finite time horizon $T = [t_0, t_f]$. Denote $X$ and $U$ as two vector-valued function sets, representing state functions and control functions respectively. Functions in $X$ (resp. $U$) are defined over $T$ and have their outputs in $\mathbb{R}^{d_x}$ (resp. $\mathbb{R}^{d_u}$). State functions $\boldsymbol{x} \in X$ and control functions $\boldsymbol{u} \in U$ are governed by a differential equation. The optimal control problem (OCP) is targeted at finding a control function that minimizes the cost functional $f$ (Lions, 1992; Kirk, 2004; Vinter & Vinter, 2010; Lewis et al., 2012):

$$\min_{\boldsymbol{u} \in U} \quad f(\boldsymbol{x}, \boldsymbol{u}) = \int_{t_0}^{t_f} p(\boldsymbol{x}(t), \boldsymbol{u}(t)) \, \mathrm{d}t + h(\boldsymbol{x}(t_f)) \tag{1a}$$

$$\text{s.t.} \quad \dot{\boldsymbol{x}}(t) = \boldsymbol{d}(\boldsymbol{x}(t), \boldsymbol{u}(t)), \tag{1b}$$

$$\boldsymbol{x}(t_0) = \boldsymbol{x}_0, \tag{1c}$$

where $\boldsymbol{d}$ is the dynamics of differential equations; $p$ evaluates the cost alongside the dynamics and $h$ evaluates the cost at the termination state $\boldsymbol{x}(t_f)$; and $\boldsymbol{x}_0$ is the initial state. We restrict our discussion to differential equation-governed optimal control problems, leaving the control problems in stochastic networks (Dai & Gluzman, 2022), inventory management (Abdolazimi et al., 2021), etc. out of the scope of this paper. The analytic solution of Eq. 1 is usually unavailable, especially for complex dynamical systems. Thus, there has been a wealth of research towards accurate, efficient, and scalable numerical OCP solvers (Rao, 2009) and neural network based solvers (Kiumarsi et al., 2017) in recent years. However, both classic and modern numerical OCP solvers are facing challenges, especially emerging in the big data era, which we briefly discuss as follows.

**1) Opacity of Dynamical Systems.** Existing works (Böhme & Frank, 2017a; Effati & Pakdaman, 2013; Jin et al., 2020) assume the dynamical systems a priori and exploit their explicit forms to ease

Table 1: Comparison of modern optimal control approaches. The proposed OptCtrlOP naturally covers all the merits in the sense of performing a single-phase direct-mapping paradigm that does not rely on known system dynamics, and supports arbitrary input-domain queries.

| Methods | Phase | Continuity | Dynamics | Reusability | Paradigm |
|---|---|---|---|---|---|
| Direct (Böhme & Frank, 2017a) | Single | Discrete | Required | No | Iterative |
| Two-Phase (Hwang et al., 2022) | Two | Discrete | Dispensable[1] | Partial[2] | Iterative |
| PDP (Jin et al., 2020) | Single | Discrete | Required | No | Iterative |
| DP (Tassa et al., 2014) | Single | Discrete | Required | No | Iterative |
| **OptCtrlOP (ours)** | Single | Continuous | Dispensable | Yes | One-pass |

[1] if phase-1 uses PINN loss (Wang et al., 2021a): required.    [2] only phase-1 is reusable.

the optimization. However, the real-world dynamical systems can be unknown and hard to model. It raises serious challenges in data collection and system inference (Schmidt et al., 2021; Ghosh et al., 2021), where special care is required for error reduction.

**2) Model Reusability.** Model reusability is conceptually measured by the capability of utilizing historical data when facing an unprecedented problem instance. Since solving an individual instance of Eq. 1 from scratch is expensive, a reusable model that can be well adapted to systems of similar forms is welcomed for practical usage. This point traces to the sensitivity analysis of optimal control (Oniki, 1973) yet is absent in recent works.

**3) Running Paradigm.** As of typical paradigms adopted in optimization, numerical optimal control solvers traditionally use iterative methods before picking the control solution, and thus introducing a multiplicative term regarding the iteration in the running time complexity. This sheds light on the high computational cost of solving a single optimal control problem.

**4) Control Solution Continuity.** Control functions are defined on a continuous domain (typically time), although being intractable for numerical solvers. Hence resorting to discretization in the input domain gives rise to the trade-off in the precision of the control solution and the computational cost, as well as the truncation errors. While the discrete solution can give point-wise queries, learning a control function for arbitrary time queries is much more valued.

**5) Running Phase.** A two-phase model (Chen et al., 2018; Wang et al., 2021a; Hwang et al., 2022) can (partially) overcome the above issues at the cost of introducing an auxiliary dynamics inference phase. This thread of works first approximates the state dynamics by a differentiable surrogate model and then, in its second phase, solves an optimization problem for control variables (more explanation in Appx. B). However, the two-phase paradigm increases computational cost and manifests inconsistency between the two phases. A motivating example in Fig. 1 shows the two-phase paradigm leads to crucial failures. When the domain of phase-2 optimization goes outside the training distribution in phase-1, this method might collapse.

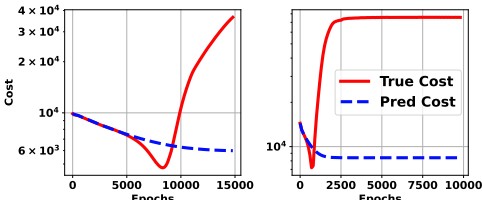

Figure 1: Phase-2 cost curves of two failed instances of two-phase control on Pendulum system. The control function gradually moves outside the training data distribution of phase-1. As a result, the control function converges w.r.t. the cost predicted by the surrogate model (blue), but diverges w.r.t. true cost (red).

Table 1 compares the methods regarding the above aspects. We propose an instance-solution operator perspective for learning to solve OCPs, thereby tackling the issues above. **The contributions are:**

1. We propose the operator perspective and solve OCPs by learning direct mappings from OCPs to their solutions. The design holds the following merits. The system dynamics is implicitly learned during the training, which relies on neither any explicit form of the system nor the optimization process at test time. As such the operator can be reused and generalized to similarly-formed OCPs without retraining, and such generalization ability is even missing for learning-free solvers. The single-phase direct mapping paradigm avoids iterative processes with substantial speedup.

2. We theoretically validate the instance-solution mapping perspective by leveraging Pontryagin's Maximum Principle, thereby converting Eq. 1 to a boundary value problem. We instantiate a neural solver: OptCtrlOP (Optimal Control OPerator), and derive bounds on its approximation error.

3. Experiments on both synthetic and real systems show that OptCtrlOP is versatile for various forms of OCPs. It achieves about 100x speedup against MLP baseline, and 10Kx speedup (on synthetic environments) over classical direct method solvers. It also generalizes well on both in- and out-of-distribution OCP instances.

**Related Works.** Most OCPs can not be solved analytically, and numerical methods are developed, which can be mainly categorized into three groups: direct methods, indirect methods, and dynamic programming. Our work bears a resemblance to indirect methods. Due to page limitation, we leave the detailed discussion on related works to Appendix B.

## 2 METHODOLOGY

In this section, we present the instance-solution operator perspective for solving OCPs. Inspired by Pontryagin's Maximum Principle (PMP) (Pontryagin, 1987), OCPs (Eq. 1) can be converted to a boundary value problem (BVP), the solution of which is a typical function operator. The operator can further be approximated accurately by neural networks with theoretical guarantees. Hence, we propose OptCtrlOP, an end-to-end OCP neural solver that learns the underlying infinite-dimensional operator $\mathcal{G}$ governed by BVP. The input of operator $\mathcal{G}$ is the cost functional $f$, and the output is the optimal control function $\boldsymbol{u}^*$, i.e. $\mathcal{G} : F \to U$. OptCtrlOP learns such an operator from paired data by minimizing the error between predicted and optimal control, without explicit knowledge of the dynamics. Our theoretical analysis guarantees that for an arbitrary small approximation error tolerance, there exists an instance of OptCtrlOP that satisfies the tolerance with bounded size and depth.

### 2.1 INSTANCE-SOLUTION OPERATOR PERSPECTIVE OF OCP

This section presents the pre-condition of our model, a novel perspective that converts OCPs into an operator that maps problem instances (defined by cost functionals and dynamics) into their solutions. The conversion has two steps: 1) convert OCP into BVP by PMP; 2) define the operator based on BVP, as explained below. Here we assume the dynamics are deterministic DE, and the derivation of stochastic DE (SDE) (Carius et al., 2022; Kishida & Ogura, 2022) is similar but based on Hamilton-Jacobi-Bellman (HJB) equation (Yong & Zhou, 1999), which is left for future work.

We elaborate how PMP converts OCP into BVP, following the derivation of the first-order necessary optimality conditions in the calculus of variations. To begin with, define the Hamiltonian $\mathcal{H}$ of Eq. 1:

$$\mathcal{H}(\boldsymbol{x}, \boldsymbol{u}, \boldsymbol{\lambda}) := p(\boldsymbol{x}, \boldsymbol{u}) + \boldsymbol{\lambda}^\top \boldsymbol{d}(\boldsymbol{x}, \boldsymbol{u}). \tag{2}$$

Then suppose $\boldsymbol{u}^* \in U$ is the optimal control function, and $\boldsymbol{x}^* \in X$ is the optimal state trajectory. The PMP asserts that there exists a co-state (adjoint) function $\boldsymbol{\lambda}^* : T \to \mathbb{R}^{d_x}$ such that the following BVP is satisfied (Pontryagin, 1987):

$$\text{dynamics:} \qquad \dot{\boldsymbol{x}}^* = \frac{\partial}{\partial \boldsymbol{\lambda}} \mathcal{H}(\boldsymbol{x}^*, \boldsymbol{u}^*, \boldsymbol{\lambda}^*), \tag{3a}$$

$$\text{co-state:} \qquad \dot{\boldsymbol{\lambda}}^* = -\frac{\partial}{\partial \boldsymbol{x}} \mathcal{H}(\boldsymbol{x}^*, \boldsymbol{u}^*, \boldsymbol{\lambda}^*), \tag{3b}$$

$$\text{optimal control:} \qquad \boldsymbol{0} = \frac{\partial}{\partial \boldsymbol{u}} \mathcal{H}(\boldsymbol{x}^*, \boldsymbol{u}^*, \boldsymbol{\lambda}^*), \tag{3c}$$

$$\text{boundary conditions:} \qquad \boldsymbol{\lambda}^*(t_f) = \frac{\partial}{\partial \boldsymbol{x}} h(\boldsymbol{x}^*(t_f)), \quad \boldsymbol{x}^*(t_0) = \boldsymbol{x}_{init}. \tag{3d}$$

A BVP is composed of a set of DEs along with boundary conditions. The BVP defines an implicit mapping from cost functional and dynamics, both of which can be represented by functions, to the optimal control function, with state and co-state being intermediate variables.

Based on BVP, one can define an operator according to concrete problem settings or protocols. In the theoretic analysis of this paper, we assume that: the initial state $\boldsymbol{x}_{init}$ and the dynamics $\boldsymbol{d} : X \times U \to \mathbb{R}$ are unknown but constant (Hwang et al., 2022). Consequently, the state $\boldsymbol{x}$ is uniquely determined by control $\boldsymbol{u}$, thus the cost functional $f : X \times U \to \mathbb{R}$ can be rewritten

interchangeably as $f_{\boldsymbol{d}} : U \to \mathbb{R}$. Under the protocol, the operator is defined as $\mathcal{G} : F \to U$, a mapping from cost functional to optimal control.

In this way, we have converted the OCPs into an operator, which directly maps OCP instances to their solutions. Theoretically, the operator is able to solve all OCPs of the same type (e.g. governed by the same dynamics $\boldsymbol{d}$), thus highly reusable and efficient. Since in practice such a non-linear operator can hardly have a closed-form expression, we propose the following neural model.

## 2.2 ARCHITECTURE OF THE PROPOSED NEURAL MODEL

From the operator perspective discussed above, constructing an OCP solver is equivalent to an operator learning problem. Inspired by DeepONet (Lu et al., 2021), we propose Operator Learning based Control Network (OptCtrlOP) to solve the OCP by learning the non-linear operator $\mathcal{G}$ directly, and theoretically estimate the error bounds. Specifically, we decompose OptCtrlOP into three components, as suggested in (Lanthaler et al., 2022):

1. **Encoder.** We define the following mapping as the encoder: $\mathcal{E} : F \to \mathbb{R}^m$, which converts the infinite-dimensional cost functional $f$ into a finite-dimensional vector $\boldsymbol{e}$ that can be processed by neural networks. For demonstration, we now let the encoder be an ad-hoc function that returns the target state $\boldsymbol{x}_{goal}$. Such a simplified encoder is still effective, since the form of the cost functional is assumed fixed, and the only parameter is the target state (Eq. 12, following Jin et al. (2020)). We term such encoder as physical priors based. Otherwise, the encoder can be implemented by functional approximations, e.g. point-wise evaluation, basis expansion, or neural network. For example, in the Heating system (Appendix E.6) the encoder output is the coefficients of a trigonometric polynomial.

2. **Approximator.** The encoded vector $\boldsymbol{e}$ is mapped to another finite dimensional vector $\boldsymbol{a}$ by the approximator mapping defined as: $\mathcal{A} : \mathbb{R}^m \to \mathbb{R}^p$.

The approximator is implemented by a neural network termed as *branch net* $\boldsymbol{\beta}$, which is a ReLU activated multi-layer perceptron (MLP) (Goodfellow et al., 2016). The output vector $\boldsymbol{a} = \boldsymbol{\beta}(\boldsymbol{e})$ will serve as the coefficient vector of basis functions produced by the following reconstructor.

3. **Reconstructor.** It reconstructs the control function $\boldsymbol{u}$ by the mapping: $\mathcal{R} : \mathbb{R}^p \to U$.

The reconstructor firstly constructs $p + 1$ basis functions via another MLP named *trunk net*, $\boldsymbol{\tau} : T \to \mathbb{R}^{p+1}$, a parametric mapping from time to basis function values. Note that the original DeepONet trunk net output is $p$-dimensional, we add an additional bias function dimension as suggested by (Lanthaler et al., 2022). Then, the control function $\boldsymbol{u} = \mathcal{R}(\boldsymbol{a})$ is reconstructed by an affine combination of basis functions, where the coefficient $\boldsymbol{a}$ is produced by the approximator:

$$\mathcal{R}(\boldsymbol{a}) := \tau_0 + \boldsymbol{a}^\top \boldsymbol{\tau}_{1:p}. \tag{4}$$

Combining three components together, OptCtrlOP maps the cost functional to a control function, $\mathcal{N} : F \to U$, and can be viewed as a composition of the above three mappings:

$$\mathcal{N} := \mathcal{R} \circ \mathcal{A} \circ \mathcal{E}. \tag{5}$$

The architecture of OptCtrlOP is summarized in Fig. 2 and the pseudo code of forward step is described Alg. 1 in Appendix A.

## 2.3 APPROXIMATION ERROR ESTIMATION

We give the estimation for the approximation error of the proposed model. The theoretic result guarantees that there exists a neural network instance of OptCtrlOP architecture (Eq. 5) approximating the operator $\mathcal{G}$ to arbitrary error tolerance. Furthermore, the size and depth of such a network are upper-bounded. The technical line of our analysis is partly inspired by (Lanthaler et al., 2022) which provides error estimation for DeepONets (Lu et al., 2021).

The error of interest in OCP is the distance between the cost of the solution and the optimal cost. Let $\mu$ be the probability measure of cost functional, with $\mu \in \mathcal{P}_2(F)$, where $\mathcal{P}_2(F)$ is the set of probability measures with finite second moments. We assume the control dimension $d_u = 1$, and $f_{\boldsymbol{d}} > 0$, w.l.o.g. Then the **approximation error** of OptCtrlOP can be defined as:

$$\widehat{\mathscr{E}} := \int_F \left| \frac{f_{\boldsymbol{d}} \circ \mathcal{N}(f) - f_{\boldsymbol{d}} \circ \mathcal{G}(f)}{f_{\boldsymbol{d}} \circ \mathcal{G}(f)} \right| \mathrm{d}\mu(f). \tag{6}$$

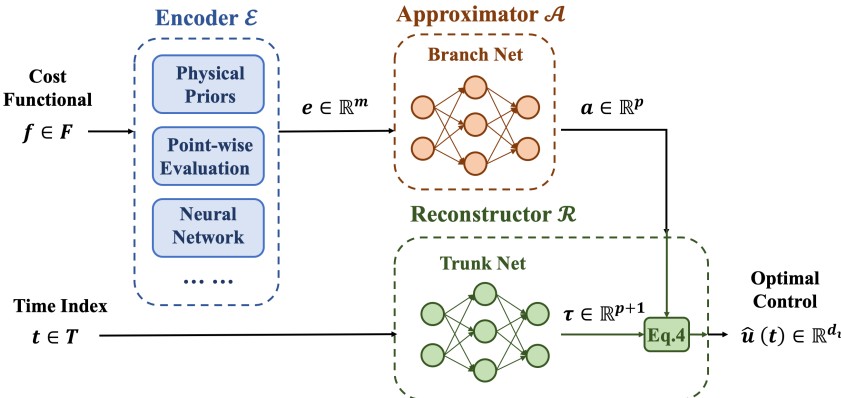

Figure 2: The architecture of OptCtrlOP. The network takes 2 inputs: cost functional $f$ and time index $t$. The input is processed by encoder, approximator and reconstructor, as introduced in Section 2.2. And finally it outputs $\hat{u}(t)$, the estimation of optimal control at time $t$.

Following the decomposition described in Sec. 2.2, the approximation error of OptCtrlOP can also be decomposed into three parts: 1) encoder error, 2) approximator error, and 3) reconstructor error.

The error of our simplified encoder is zero, since there exists a one-to-one mapping between the target state (encoder output) and the cost functional (encoder input). In other words, for a given OptCtrlOP encoder $\mathcal{E}$, there exists an inverse mapping $\mathcal{E}^{-1}$, such that $\mathcal{E}^{-1} \circ \mathcal{E} = \text{Id}$. If the encoder is implemented by functional approximations, e.g. point-wise evaluation (Lu et al., 2021), then the encoder error should be considered (Appx D).

For a reconstructor $\mathcal{R}$, its error $\widehat{\mathscr{E}}_{\mathcal{R}}$ is estimated by the mismatch between $\mathcal{R}$ and its approximate inverse mapping, projector $\mathcal{P}$, weighted by push-forward measure $\mathcal{G}_{\#}\mu(u) := \mu(\mathcal{G}^{-1}(u))$.

$$
\begin{aligned}
\widehat{\mathscr{E}}_{\mathcal{R}} &:= \left( \int_U \| \mathcal{R} \circ \mathcal{P}(u) - u \|_U \, \mathrm{d}\left(\mathcal{G}_{\#}\mu\right)(u) \right)^{\frac{1}{2}} \\
\mathcal{P} &:= \operatorname*{argmin}_{\mathcal{P}} \widehat{\mathscr{E}}_{\mathcal{R}}, \quad \text{s.t.} \quad \mathcal{P} \circ \mathcal{R} = \text{Id}.
\end{aligned}
\tag{7}
$$

Intuitively, such reconstructor error quantifies the information loss induced by $\mathcal{R}$. An ideal $\mathcal{R}$ without any information loss should be invertible, i.e. its optimal inverse $\mathcal{P}$ is exactly $\mathcal{R}^{-1}$, thus we have $\widehat{\mathscr{E}}_{\mathcal{R}} = 0$.

Given the encoder and reconstructor, and denote the encoder output is $e \in \mathbb{R}^m$, the error $\widehat{\mathscr{E}}_{\mathcal{A}}$ induced by approximator $\mathcal{A}$ is defined as the distance between the approximator output and the optimal coefficient vector, weighted on push-forward measure $\mathcal{E}_{\#}\mu(e) := \mu(\mathcal{E}^{-1}(e))$:

$$
\widehat{\mathscr{E}}_{\mathcal{A}} := \left( \int_{\mathbb{R}^m} \| \mathcal{A}(e) - \mathcal{P} \circ \mathcal{G} \circ \mathcal{E}^{-1}(e) \|_{\ell^2(\mathbb{R}^p)}^2 \, \mathrm{d}\left(\mathcal{E}_{\#}\mu\right)(e) \right)^{\frac{1}{2}}.
\tag{8}
$$

With the definitions above, we can estimate the approximation error of our OptCtrlOP by the error of each of its components, as stated in the following theorem (see detailed proof in Appendix C.1):

**Theorem 1** (**Decomposition of OptCtrlOP Approximation Error**). *Suppose the cost functional $f_d$ is Lipschitz continuous, with Lipschitz constant $\text{Lip}(f_d)$. Define constant $C = \sup_{f \in F} \frac{\text{Lip}(f_d)}{f_d \circ \mathcal{G}(f)}$. The approximation error $\widehat{\mathscr{E}}$ (Eq. 6) of a OptCtrlOP $\mathcal{N} = \mathcal{R} \circ \mathcal{A} \circ \mathcal{E}$ is upper-bounded by*

$$
\widehat{\mathscr{E}} \leq C \left( \text{Lip}(\mathcal{R}) \widehat{\mathscr{E}}_{\mathcal{A}} + \widehat{\mathscr{E}}_{\mathcal{R}} \right).
\tag{9}
$$

The estimation of reconstructor error $\widehat{\mathscr{E}}_{\mathcal{R}}$ is analyzed in the previous work (Lanthaler et al., 2022), which gives a detailed discussion of the error estimation of DeepONet, and the reconstructor error component is the same as that of OptCtrlOP. We cite the result to establish the following theorem:

**Theorem 2** (**Reconstructor Error** (Lanthaler et al., 2022))**.** *If $\mathcal{G}$ defines a Lipschitz mapping $\mathcal{G}$ : $F \to H^s(T)$ ,for some $s > 0, M > 0$, with $\int_F \|\mathcal{G}(f)\|^2_{H^s} \mathrm{d}\mu(f) \leq M < \infty$, then there exists a constant $C = C(s, M) > 0$, such that for any $p \in \mathbb{N}$, there exists a trunk net $\boldsymbol{\tau} : T \to \mathbb{R}^{p+1}$ (with bias term $\tau_0 \equiv 0$ ) and the associated reconstruction $\mathcal{R} : \mathbb{R}^p \to U$ satisfies:*

$$\text{size}(\boldsymbol{\tau}) \leq Cp\left(1 + \log(p)^2\right), \qquad \text{depth}(\boldsymbol{\tau}) \leq C\left(1 + \log(p)^2\right), \qquad \widehat{\mathscr{e}}_{\mathcal{R}} \leq Cp^{-s}. \tag{10}$$

*Furthermore, the reconstruction $\mathcal{R}$ and the optimal projection $\mathcal{P}$ satisfy $\text{Lip}(\mathcal{R}), \text{Lip}(\mathcal{P}) \leq 2$.*

Note that $\text{size}(\cdot)$ is defined as the number of trainable parameters of a neural network, and $\text{depth}(\cdot)$ denotes the number of hidden layers. $H^s$ is a Sobolev space with $s$ degrees of regularity and $L^2$ norm. The proof (omitted here) is based an observation that reconstructor is minimized when trunk net outputs $\{\tau_1, \tau_2, ..., \tau_p\}$ are linearly independent. The observation is consistent with intuition, since trunk net outputs are regarded as basis functions (Eq. 4).

Next, the error bound of the approximator will be presented. Recall that the approximator learns a mapping between vector space by MLP, whose error estimation is well studied in deep learning theory. One of the existing works (Gühring et al., 2020) derives the estimation based on the Sobolev regularity of the mapping. We extend their result to form the following theorem (proof given in Appendix C.2):

**Theorem 3** (**Approximator Error**)**.** *Given operator $\mathcal{G} : F \to U$, encoder $F \to \mathbb{R}^m$, and reconstructor $\mathcal{R} : \mathbb{R}^p \to U$, let $\mathcal{P}$ denote the corresponding projector. If for some $s \in \mathbb{N}_{\geq 2}, M > 0$, the following bound is satisfied: $\left\|\mathcal{P} \circ \mathcal{G} \circ \mathcal{E}^{-1}\right\|_{H^s(\mathcal{E}_\# \mu)} \leq M < \infty$, then there exists a constant $C = C(m, s, M) > 0$, such that for any $\varepsilon \in (0, \frac{1}{2})$, there exists an approximator $\mathcal{A} : \mathbb{R}^m \to \mathbb{R}^p$ :*

$$\text{size}(\mathcal{A}) \leq Cp^2\varepsilon^{-m/s}\log\left(\varepsilon^{-1}\right), \qquad \text{depth}(\mathcal{A}) \leq C\log\left(\varepsilon^{-1}\right), \qquad \widehat{\mathscr{e}}_{\mathcal{A}} \leq \sqrt{p}\varepsilon. \tag{11}$$

In summary, we have proved that the approximation error is bounded by the sum of the reconstructor and approximator errors. Those errors can hold under arbitrary small tolerance, with bounded size and depth. The theorems hold as long as some integratable and continuous conditions are satisfied, which is trivial in real-world OCPs.

## 3 EXPERIMENTS

### 3.1 SYNTHETIC CONTROL SYSTEMS

#### 3.1.1 CONTROL SYSTEMS AND DATA GENERATION

We evaluate OptCtrlOP on six representative optimal control systems by following the same protocol of (Jin et al., 2020; Hwang et al., 2022), as summarized in Table 4. We postpone the rest systems to Appendix E, and only describe the details of the Quadrotor control here:

$$\dot{\boldsymbol{p}} = \boldsymbol{v}, \qquad m\dot{\boldsymbol{v}} = \begin{bmatrix} 0 \\ 0 \\ mg \end{bmatrix} + \boldsymbol{R}^\top(\boldsymbol{q}) \begin{bmatrix} 0 \\ 0 \\ \mathbf{1}^\top\boldsymbol{u} \end{bmatrix}, \qquad \dot{\boldsymbol{q}} = \frac{1}{2}\boldsymbol{\Omega}(\boldsymbol{\omega})\boldsymbol{q}, \qquad \boldsymbol{J}\dot{\boldsymbol{\omega}} = \boldsymbol{T}\boldsymbol{u} - \boldsymbol{\omega} \times \boldsymbol{J}\boldsymbol{\omega}.$$

This system describes the dynamics of a helicopter with four rotors. The state $\boldsymbol{x} = [\boldsymbol{p}^\top, \boldsymbol{v}^\top, \boldsymbol{\omega}^\top]^\top \in \mathbb{R}^9$ consists of parts: position $\boldsymbol{p}$, velocity $\boldsymbol{v}$, and angular velocity $\boldsymbol{\omega}$. The control $\boldsymbol{u} \in \mathbb{R}^4$ is the thrusts of the four rotating propellers of the quadrotor. $\boldsymbol{q} \in \mathbb{R}^4$ is the unit quaternion (Jia, 2019) representing the attitude(spacial rotation) of quadrotor w.r.t. the inertial frame. $\boldsymbol{J}$ is the moment of inertia in quadrotor's frame, and $\boldsymbol{T}\boldsymbol{u}$ is the torque applied to the quadrotor. Our setting is similar to (Jin et al., 2020, Appx. E.1), but we exclude the quaternion from the state. We set the initial state $\boldsymbol{x}_{init} = [-8, -6, 9, \mathbf{0}]^\top$, the initial quaternion $\boldsymbol{q}_{init} = \mathbf{0}$. The matrices $\boldsymbol{\Omega}(\boldsymbol{\omega}), \boldsymbol{R}(\boldsymbol{q}), \boldsymbol{T}$ are coefficient matrices, see definition in Appx. E.4. The cost functional is defined as following, with coefficients $\boldsymbol{c}_{\boldsymbol{x}} = \mathbf{1}, c_u = 0.1$.

$$\min_u \quad \int_0^{tf} \boldsymbol{c}_{\boldsymbol{x}}^\top(\boldsymbol{x}(t) - \boldsymbol{x}_{goal})^2 + c_u\|\boldsymbol{u}(t)\|^2 \, \mathrm{d}t \tag{12}$$

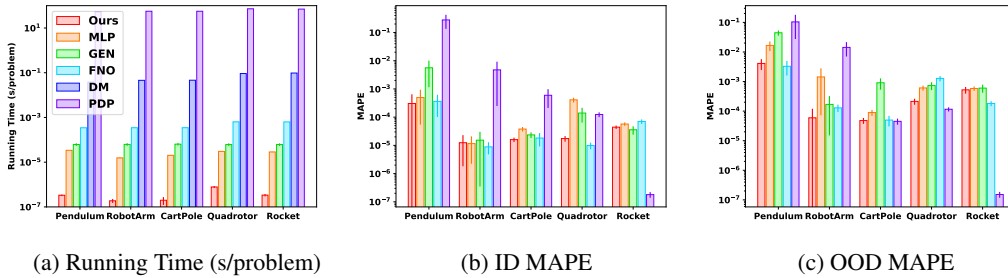

(a) Running Time (s/problem)    (b) ID MAPE    (c) OOD MAPE

Figure 3: The running time and mean absolute percentage error (MAPE) on in-distribution (ID) and out-of-distribution (OOD) benchmark problem sets. Compared with baselines, OptCtrlOP (red bars) achieves higher or comparable accuracy, with over 100x speedup.

With the settings above given, the solution of OCP only depends on the target state $x_{goal}$. Therefore, we generate datasets (for model training/validation) and benchmarks (for model testing) by sampling target states from a pre-defined distribution. To fully evaluate the generalization ability, we define both in-distribution (ID) and out-of-distribution (OOD) (Shen et al., 2021). Specifically, we design two random variables, $\mathbf{x}_{goal}^{in} := x_{goal}^{base} + \epsilon_{in}$, and $\mathbf{x}_{goal}^{out} := x_{goal}^{base} + \epsilon_{out}$, where $x_{goal}^{base}$ is a baseline goal state, and $\epsilon_{in,out}$ are different noise applied to ID and OOD. In Quadrotor problems for example, we set $x_{goal}^{base} = \mathbf{0}$, and uniform noise $\epsilon_{in} \sim \mathcal{U}(0.1, 1.1)$, and $\epsilon_{out} \sim \mathcal{U}(-0.1, 0.1)$.

The training data are sampled from ID, while validation data and benchmark sets are both sampled from ID and OOD separately. The data generation process is shown in Alg. 2 in Appendix. For a given distribution, we sample a target state $x_{goal}$, and construct the corresponding cost functional $f$ and OCP. Then define 100 time indices uniformly spaced in time horizon $T = [0, \mathrm{tf}]$, $\mathrm{tf} \sim \mathcal{U}(1, 1.01)$. The length of $T$ is slightly perturbed so that time indices fall in the whole horizon, instead of fixed points. Then we solve the resulting OCP by the DM solver, and get the optimal control $u^*$ at those time indices. After that we sample 10 time indices $\{t_i\}_{1 \le i \le 10}$, creating 10 triplets $\{(f, t_i, u^*(t_i))\}_{1 \le i \le 10}$ and adding them to the dataset. Repeat the process, until the size meets the requirement. The benchmark set is generated in the same way, but we store $(f, J_{opt})$ pair ($J_{opt}$ is the optimal cost) for each problem instead.

### 3.1.2 IMPLEMENTATION AND BASELINES

For all systems and all neural models, the learning rate starts from 0.01, decaying every 1,000 epochs at a rate of 0.9. The batch size is 10,000, and the optimizer is Adam (Kingma & Ba, 2014). The loss is the mean squared error defined below, with a dataset $D$ of $N$ samples: $\mathcal{L} = \frac{1}{N} \sum_{i,j \in D} \left\| \mathcal{N}(x_{goal,j})(t_i) - u_j^*(t_i) \right\|^2$. For comparison, we choose the following baselines. Other details of implementation and baselines are recorded in Appendix F.

1) **Direct Method (DM):** a classical direct OCP solver, with Interior Point OPTimizer (IPOPT) (Biegler & Zavala, 2009) backend NLP solver.

2) **Pontryagin Differentiable Programming (PDP)** (Jin et al., 2020): an adjoint-based indirect method, differentiating through PMP, and optimized by gradient descent.

3) **Multi-layer Perceptron (MLP)**: a fully connected counterpart (Alg. 4) of OptCtrlOP.

4) **Fourier Neural Operator (FNO)** (Li et al., 2020): A neural operator consists of consecutive Fourier transform layers.

5) **Graph Element Network (GEN)** (Alet et al., 2019): A graph neural operator with graph convolution backbone.

### 3.1.3 RESULTS AND DISCUSSION

We present the numerical results on the six systems to evaluate the efficiency and accuracy of OptCtrlOP. The metrics of interest are 1) the running time of solving problems; 2) the quality of solution, measured by mean absolute percentage error (MAPE) between the true optimal cost and the predicted cost, which is defined as the mean of $|(J_{opt} - J_{sol})/J_{opt}|$, where $J_{opt}$ is the optimal cost generated by DM (regarded as ground truth), and $J_{sol}$ is the cost of the solution produced by the

Table 2: Performance of Quadrotor environment.

| Model | Time (sec./instance) | ID MAPE | OOD MAPE | ID-OOD Gap |
|---|---|---|---|---|
| DM (Böhme & Frank, 2017a) | $9.23 \times 10^{-2}$ | $\diagdown$ | $\diagdown$ | $\diagdown$ |
| OptCtrlOP | $7.79 \times 10^{-7}$ | $1.74 \times 10^{-5}$ | $2.13 \times 10^{-4}$ | $1.96 \times 10^{-4}$ |
| MLP (Alg. 4) | $3.04 \times 10^{-5}$ | $4.12 \times 10^{-4}$ | $6.09 \times 10^{-4}$ | $1.97 \times 10^{-4}$ |
| GEN (Alet et al., 2019) | $6.15 \times 10^{-5}$ | $1.40 \times 10^{-4}$ | $7.31 \times 10^{-4}$ | $5.91 \times 10^{-4}$ |
| FNO (Li et al., 2020) | $6.38 \times 10^{-4}$ | $9.87 \times 10^{-6}$ | $1.28 \times 10^{-3}$ | $1.27 \times 10^{-3}$ |
| PDP (Jin et al., 2020) | $7.25 \times 10^{1}$ | $1.24 \times 10^{-4}$ | $1.16 \times 10^{-4}$ | $8.74 \times 10^{-6}$ |

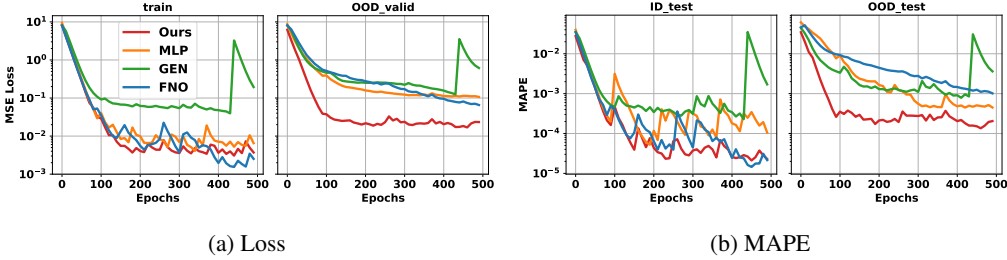

(a) Loss                                        (b) MAPE

Figure 4: (a) Quadrotor Loss curve. (b) Quadrotor cost MAPE (mean absolute percentage error) curve. For ID (train and ID_test), OptCtrlOP (red curves) performs competitively, or better than others. It also outperforms all neural baselines on OOD.

model. The MAPE is calculated on ID/OOD benchmarks respectively, and the running time is averaged for 2,000 random problems. The results of ODE-constrained OCP are visualized in Fig. 3.

First, the comparison of running time is shown in Fig. 3a, which shows that the neural operator solver is much faster than the classic solver. For example, OptCtrlOP achieves over $10^5$ times speedup against the DM solver. The acceleration can be reasoned in two aspects: 1) the neural operator solvers produce the output by a single forward propagation, while the classic methods need to iterate between forward and backward pass multiple times; 2) the neural solver calculation is highly paralleled. Moreover, OptCtrlOP is 100 times faster than MLP, although both of them are neural operator models. The difference is rooted in OptCtrlOP architecture, where two inputs (i.e. cost functional and time index) are processed by branch net and trunk net separately. Consequently, OptCtrlOP runs $\mathcal{O}(M + N)$ forward passes, if the benchmark set consists of $N$ problems, and the time horizon is discretized into $M$ indices. In comparison, MLP entangles two inputs together, requiring $\mathcal{O}(MN)$ forward passes. The difference is also given in Alg. 3-4 in Appendix. The FNO follows a similar diagram as MLP, but 10+ times slower than MLP, since it involves computationally intensive Fourier transformations. The GEN processes two inputs in a disentangled manner, similar to that of OptCtrlOP. But GEN is 100+ times slower than OptCtrlOP, probably because of the intrinsic complexity of graph construction and graph convolution.

Next, the accuracy on in- and out-of-distribution benchmark sets is compared in Fig. 3b-3c. Compared with other neural models, OptCtrlOP achieves better or comparable accuracy in general. In addition, OptCtrlOP outperforms classical PDP on more than half of the benchmarks. As a concrete example, we investigate the performance of Quadrotor environment. From Tab. 2 (and Fig. 4 for learning trajectory of neural models), one can observe that MAPE of OptCtrlOP on ID is the second lowest among all models, with a slight disadvantage compared to FNO. On OOD, however, OptCtrlOP outperforms FNO by a clear margin, and the performance is close to that of the classical method PDP. Among all neural models, OptCtrlOP achieves the lowest OOD MAPE as well as the smallest ID-OOD gap (defined as the absolute distance between ID and OOD MAPE). We conjecture that the OOD generalization ability of OptCtrlOP results from its architecture, where branch net and trunk net (coefficients and basis functions) are explicitly disentangled. Such a structure may inherit the inductive bias from numerical basis expansion methods (e.g. Kafash et al. (2014)), thus being more robust to distribution shifts. Due to space limitations, the numerical results of other systems are given in Tab. 6-10 in Appendix.

## 3.2 REAL-WORLD CONTROL DATASET OF PLANAR PUSHING

Table 3: Results of Pushing environment.

| Model | Time(sec./instance) | ID MAPE | OOD MAPE |
|---|---|---|---|
| OptCtrlOP | $6.07 \times 10^{-6}$ | 0.109 | 0.136 |
| MLP | $8.85 \times 10^{-4}$ | 0.128 | 0.149 |
| GEN | $6.59 \times 10^{-5}$ | 0.201 | 0.207 |
| FNO | $1.34 \times 10^{-3}$ | 0.135 | 0.144 |

This section presents how to learn optimal control of a robot arm for pushing objects of varying shapes on various planar surfaces. We use the *Pushing* dataset Yu et al. (2016), a challenging dataset consisting of noisy, real-world data produced by an ABB IRB 120 industrial robotic arm (Fig.5, right part). The robot arm is controlled to push objects starting from various contact positions and 9 angles (initial state), along different trajectories (target state functions), with 11 object shapes and 4 surface materials (dynamics). The control function is represented by the force exerted on to object, measured by the force sensor. Left of Fig.5 gives a compact overview of input variables, and more details are given in Appx. E.8.

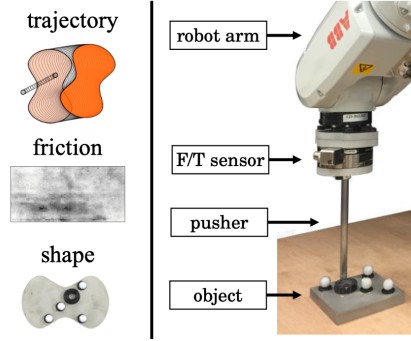

Figure 5: Pushing environment, composed of images from Yu et al. (2016).

In our experiment, we apply OptCtrlOP to learn a mapping from a pushing OCP instance (represented by variables above) to the optimal control function. The input now is no longer cost functional $f$ only, but $f_d$ with abuse of notation, where subscript $d$ denotes dynamics and initial conditions. And the encoder is realized by different techniques for different inputs, such as Savitzky–Golay smoothing (Savitzky & Golay, 1964) and down-sampling for trajectories, mean and standard value extraction for friction map, and Convolution Network(CNN) (LeCun et al., 1989) for shape images. The encoder error now is not negligible, but the analysis framework of approximation error can be extended to include this error (Appx. D). The estimation of encoder error itself depends on the specific choice of the encoder, which is beyond the scope of this paper and is left to future work.

We extract training data from ID, validation and test data from both ID/OOD. The ID/OOD is distinguished by different initial contact positions. The accuracy metric MAPE is now defined as $\|\hat{u} - u^*\|/\|u^*\|$. We compare OptCtrlOP performance only with neural baselines, since the explicit expression of pushing OCP is unavailable. All neural models share the same encoder structure, while the parameters of CNN are trained end-to-end individually for each model. The results are displayed in Tab. 3, from which one can observe OptCtrlOP outperforms all baselines in both running time and ID/OOD accuracy. Notice that the performance of FNO and GEN degrades compared with that of synthetic data. The reason might be that their architecture is not suitable for complex OCP tasks like pushing. For example, the essence of FNO is to learn parametric transformation in Fourier space. For the pushing dataset, however, the input OCP instance is a composition of several functions and environment parameters, of which the Fourier transform does not have a clear physical meaning.

## 4 Conclusion

**Future Works.** This paper studies an effective way to solve general OCPs with data-driven approaches. We do not specify the forms of problem instances or investigate sophisticated models for specific problems, which calls for careful designs and exploitation of the problem structures. We also leave rigorous analysis on OOD performance and encoder error as future works.

**Conclusion.** We have proposed a novel instance-solution operator perspective of OCPs, where the operator directly maps cost functionals to optimal control functions. Based on this perspective, we present a neural operator OptCtrlOP, with a theoretic guarantee on its approximation capability. Extensive experiments on various OCP system benchmarks show the outstanding generalization ability and efficiency of OptCtrlOP, on both ID and OOD settings. We envision the proposed model will be beneficial in solving numerous high-dimensional problems in the learning and control fields.

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

## A    ALGORITHMS

---

**Algorithm 1:** OptCtrlOP forward step (single sample version)

---

**Input:** cost functional $f$, and time index $t_i$
**Output:** estimated control $\hat{\boldsymbol{u}}(t_i) = \mathcal{N}(f)(t_i)$
1  $\boldsymbol{e} \leftarrow \mathcal{E}(f)$ ; `// Encoder`
2  $\boldsymbol{a} \leftarrow \mathcal{A}(\boldsymbol{e})$ ; `// Approximator`
3  $\boldsymbol{r} \leftarrow \boldsymbol{\tau}(t_i)$; `// trunk net (Reconstructor step 1)`
4  $\hat{\boldsymbol{u}}(t_i) \leftarrow \boldsymbol{r}_0 + \boldsymbol{a}^\top \boldsymbol{r}_{1:p}$; `// affine combination (Reconstructor step 2)`
5  **return** $\hat{\boldsymbol{u}}(t_i)$

---

---

**Algorithm 2:** Data generation

---

**Input:** Distribution of target state $\mathbf{x}_{goal}$
**Output:** Dataset
1  $N \leftarrow$ Number of samples per trajectory;
2  Dataset $\leftarrow$ empty set;
3  **while** *not Dataset is full* **do**
4      sample $\boldsymbol{x}_{goal}$ from $\mathbf{x}_{goal}$;
5      construct cost functional $f$ (e.g. Eq. 12) with $\boldsymbol{x}_{goal}$;
6      construct OCP (Eq. 1) with cost functional $f$;
7      $\boldsymbol{u}^* \leftarrow$ OC_Solver(OCP); `// Any solver is applicable.  We choose DM.`
8      sample $\{t_i\}_{1 \leq i \leq N}$ from time horizon $T$ of OCP;
9      add triplets $\{(f, t_i, \boldsymbol{u}^*(t_i))\}_{1 \leq i \leq N}$ to Dataset;
10 **return** *Dataset*

---

---

**Algorithm 3:** OptCtrlOP inference on benchmark set

---

**Input:** Benchmark set of cost functionals $F = \{f_j\}_{j \leq N}$, time indices $\boldsymbol{t} = \{t_i\}_{i \leq M}$
**Output:** estimated controls $\hat{\boldsymbol{U}} = \{\hat{\boldsymbol{u}}_j(t_i)\}_{i \leq M, j \leq N}$
1  $\boldsymbol{E} \leftarrow \mathcal{E}(F)$ ; `// Encoder`
2  $\boldsymbol{A} \leftarrow \mathcal{A}(\boldsymbol{E})$ ; `// Approximator, ` $\mathcal{O}(N)$
3  $\boldsymbol{R} \leftarrow \boldsymbol{\tau}(\boldsymbol{t})$; `// trunk net (Reconstructor step 1), ` $\mathcal{O}(M)$
4  **for** $j \leftarrow 1$ **to** $N$ **do**
5      $\hat{\boldsymbol{u}}_j(\boldsymbol{t}) \leftarrow \boldsymbol{R}_0 + \boldsymbol{A}_j^\top \boldsymbol{R}_{1:p}$; `// affine combination (Reconstructor step 2)`
6  **return** $\hat{\boldsymbol{U}}$

---

## B    BACKGROUND AND RELATED WORK

### B.1    OCP SOLVERS

Traditional numerical solvers are well developed over the decades, which are learning-free and often involve tedious optimization iterations to find an optimal solution.

**Direct methods** (Böhme & Frank, 2017a) reformulate OCP as finite-dimensional nonlinear programming (NLP) (Bazaraa et al., 2013), and solve the problem by NLP algorithms, e.g. sequential quadratic programming (Boggs & Tolle, 1995) and interior-point method (Mehrotra, 1992). The reformulation essentially constructs surrogate models, where the state and control function (infinite dimension) is replaced by polynomial or piece-wise constant functions. The dynamics constraint is discretized into equality constraints. The direct methods optimize the surrogate models, thus the solution is not guaranteed to be optimal for the origin problem. Likewise, typical direct neural solvers (Chen et al., 2018; Wang et al., 2021a; Hwang et al., 2022), termed as Two-Phase models, consist of two phases: 1) approximating the dynamics by a neural network (surrogate model); 2) solving the NLP via gradient descent, by differentiating through the network. The advantage of Two-Phase

---

**Algorithm 4:** MLP inference on benchmark set

---

**Input:** Benchmark set of cost functionals $F = \{f_j\}_{j \leq N}$, time indices $\boldsymbol{t} = \{t_i\}_{i \leq M}$
**Output:** estimated controls $\hat{\boldsymbol{U}} = \{\hat{\boldsymbol{u}}_j(t_i)\}_{i \leq M, j \leq N}$
1 **for** $i \leftarrow 1$ **to** $M$ **do**
2     **for** $j \leftarrow 1$ **to** $N$ **do**
3        $\boldsymbol{e}_j \leftarrow \mathcal{E}(f_j)$; // Encoder
4        $\hat{\boldsymbol{u}}_j(t_i) \leftarrow \text{MLP}(\boldsymbol{e}_j, t_i)$; // repeat $\mathcal{O}(MN)$ times

5 **return** $\hat{U}$

---

models against traditional direct methods is computational efficiency, especially in high-dimensional cases. However, the two-phase method is sensitive to distribution shift (see Fig. 1).

**Indirect methods** (Böhme & Frank, 2017b) are based on Pontryagin's Maximum Principle (PMP) (Pontryagin, 1987). By PMP, indirect methods convert OCP (Eq. 1) into a boundary-value problem (BVP) (Lasota, 1968), which is then solved by numerical methods such as shooting method (Bock & Plitt, 1984), collocation method (Xiu & Hesthaven, 2005), adjoint-based gradient descend (Effati & Pakdaman, 2013; Jin et al., 2020). These numerical methods are sensitive to the initial guesses of the solution. Some indirect methods based neural solvers approximate the finite-dimensional mapping from state $\boldsymbol{x}^*(t) \in \mathbb{R}^{d_x}$ to control $\boldsymbol{u}^*(t) \in \mathbb{R}^{d_u}$ (Cheng et al., 2020), or to co-state $\boldsymbol{\lambda}^*(t) \in \mathbb{R}^{d_x}$ (Xie et al., 2018). The full trajectory of the control function is obtained by repeatedly applying the mapping and getting feedback from the system, and such sequential nature is the efficiency bottleneck. Another work (D'ambrosio et al., 2021) proposes to solve the BVP via a PINN, thus its trained network works only for one specific OCP instance. Distinctive from all these models, OptCtrlOP solves BVP by learning an infinite-dimensional operator that maps cost functional $f \in F$ to control function $\boldsymbol{u}^* \in U$. The trained model is available for parallel queries on different OCP instances, with high efficiency and accuracy.

**Dynamic programming (DP)** is an alternative, based on Bellman's principle of optimality (Bellman & Kalaba, 1960). It offers a rule to divide a high-dimensional optimization problem with a long time horizon into smaller, easier-to-solve auxiliary optimization problems. Typical methods are: Hamilton-Jacobi-Bellman (HJB) equation (Al-Tamimi et al., 2008), differential dynamical programming (DDP) (Tassa et al., 2014), which assumes quadratic dynamics and value function, and iterative linear quadratic regulator (iLQR) (Li & Todorov, 2004), which assumes linear dynamics and quadratic value function. Similar to dynamic programming, model predictive control (MPC) synthesizes the approximate control function via the repeated solution of finite overlapping horizons (Hewing et al., 2020). The main drawback of DP is the *curse of dimensionality* on the number and complexity of the auxiliary problem. MPC alleviates this problem at the expense of optimality. Yet fast implementation of MPC is still under exploration and remains open (Nubert et al., 2020).

### B.2 DIFFERENTIAL EQUATION NEURAL SOLVERS

A variety of networks have been developed to solve DE, including Physics-informed neural networks (PINNs) (Raissi et al., 2019), neural operators (Lu et al., 2021), hybrid models (Mathiesen et al., 2022; Lienen & Günnemann, 2021), and frequency domain models (Li et al., 2020; Poli et al., 2022). We will briefly introduce the first two models for their close relevance to our work.

**PINNs** parameterize the DE solution as a neural network, and learn the parameters by minimizing the residual loss and boundary condition loss (Yu et al., 2018; Raissi et al., 2019; Sirignano & Spiliopoulos, 2018). PINNs are similar to those numerical methods e.g. the finite element method in that they replace the linear span of a finite set of local basis functions with neural networks. PINNs usually have simple architectures (e.g. MLP), although they have produced remarkable results across a wide range of areas in computational science and engineering (Raissi et al., 2020; Zhu et al., 2019). However, these models are limited to learning the solution of one specific DE instance, but not the operator. In other words, if the coefficients of the DE instance slightly change, then a new neural network needs to be trained accordingly, which is time-consuming. Another major drawback of PINNs, as pointed out by (Wang et al., 2021b), is that the magnitude of two loss terms (i.e.residual loss and boundary condition loss) is inherently imbalanced, leading to heavily biased predictions even for simple linear equations.

**Neural operators** regards DE as an operator that maps the input to the solution. Learning operators using neural networks was introduced in the seminal work (Chen & Chen, 1995). It proposes the universal approximation theorem for operator learning, i.e. a network with a single hidden layer can approximate any nonlinear continuous operator. Lu et al. (2021) follows this theorem by designing a deep architecture named *DeepONet*, which consists of two networks: a branch net for input functions and a trunk net for the querying locations in the output space. We choose this architecture as our OCP operator learner, and our analysis of optimal control error bound is partly inspired by Lanthaler et al. (2022), providing error estimation of DeepONet. Note that DeepONet is designed to solve DE operators, and is not ready to handle optimization tasks e.g. OC.

In addition, there exist many neural operators with other architectures. For example, another type of neural operator is to parameterize the operator as a convolutional neural network (CNN) between the finite-dimensional data meshes (Guo et al., 2016; Zhu & Zabaras, 2018; Khoo et al., 2021). The major weakness of these models is that it is impossible to query solutions at off-grid points. Moreover, graph neural networks (GNNs) (Kipf & Welling, 2016) are also applied in operator learning (Alet et al., 2019; Anandkumar et al., 2020). The key idea is to construct the spacial mesh as a graph, where the nodes are discretized spatial locations, and the edges link neighboring locations. Compared with CNN-based models, the graph operator model is less sensitive to resolution, and is capable of inferencing at off-grid points by adding new nodes to the graph. However, its computational cost is still high, growing quadratically with the number of nodes. Another category of neural operators is Fourier transform based (Li et al., 2020; Kovachki et al., 2021). The models learn parametric linear functions in the frequency domain, along with nonlinear functions in the time domain. The conversion between those two domains is realized by discrete Fourier transformation.

## C PROOFS

### C.1 PROOF OF THEOREM 1

*Proof.* Firstly, extract the constant, and decompose the error by triangle inequality (subscript of norm omitted):

$$
\begin{aligned}
\widehat{\mathscr{E}} &= \left| \frac{f_{\boldsymbol{d}} \circ \mathcal{G} - f_{\boldsymbol{d}} \circ \mathcal{N}}{f_{\boldsymbol{d}} \circ \mathcal{G}} \right| \\
&\leq \sup_{f \in F} \left( \frac{\mathrm{Lip}(f_{\boldsymbol{d}})}{f_{\boldsymbol{d}} \circ \mathcal{G}} \right) \|\mathcal{G} - \mathcal{N}\| = C\|\mathcal{G} - \mathcal{N}\| \\
\|\mathcal{G} - \mathcal{N}\| &= \|\mathcal{G} - \mathcal{R} \circ \mathcal{A} \circ \mathcal{E}\| \\
&= \|\mathcal{G} - \mathcal{R} \circ \mathcal{P} \circ \mathcal{G} + \mathcal{R} \circ \mathcal{P} \circ \mathcal{G} - \mathcal{R} \circ \mathcal{A} \circ \mathcal{E}\| \\
&\leq \|\mathcal{G} - \mathcal{R} \circ \mathcal{P} \circ \mathcal{G}\| + \|\mathcal{R} \circ \mathcal{P} \circ \mathcal{G} - \mathcal{R} \circ \mathcal{A} \circ \mathcal{E}\|
\end{aligned}
$$

The first term is exactly the reconstructor error $\widehat{\mathscr{E}}_{\mathcal{R}}$, by definition of push-forward:

$$
\|\mathcal{G} - \mathcal{R} \circ \mathcal{P} \circ \mathcal{G}\|_{L^2(\mu)} = \|\mathrm{Id} - \mathcal{R} \circ \mathcal{P}\|_{L^2(\mathcal{G}_{\#}\mu)} = \widehat{\mathscr{E}}_{\mathcal{R}}
$$

And the second term is related to the approximator error $\widehat{\mathscr{E}}_{\mathcal{A}}$:

$$
\begin{aligned}
\|\mathcal{R} \circ \mathcal{P} \circ \mathcal{G} - \mathcal{R} \circ \mathcal{A} \circ \mathcal{E}\|_{L^2(\mu)} &= \left\|\mathcal{R} \circ \mathcal{P} \circ \mathcal{G} \circ \mathcal{E}^{-1} \circ \mathcal{E} - \mathcal{R} \circ \mathcal{A} \circ \mathcal{E}\right\|_{L^2(\mu)} \\
&\leq \mathrm{Lip}(\mathcal{R})\left\|\mathcal{P} \circ \mathcal{G} \circ \mathcal{E}^{-1} \circ \mathcal{E} - \mathcal{A} \circ \mathcal{E}\right\|_{L^2(\mu)} \\
&= \mathrm{Lip}(\mathcal{R})\left\|\mathcal{P} \circ \mathcal{G} \circ \mathcal{E}^{-1} \circ \mathcal{E} - \mathcal{A} \circ \mathcal{E}\right\|_{L^2(\mathcal{E}_{\#}\mu)} \\
&= \mathrm{Lip}(\mathcal{R})\widehat{\mathscr{E}}_{\mathcal{A}}
\end{aligned}
$$

$\square$

### C.2 PROOF OF THEOREM 3

*Proof.* Our estimation of approximator error is based on the approximation rates of deep ReLU neural networks derived in Gühring et al. (2020) (notation modified for consistency):

**Theorem 4.** *Let $m \in \mathbb{N}, s \in \mathbb{N}_{\geq 2}, 1 \leq q \leq \infty, M > 0$, and $0 \leq n \leq 1$, then there exists a constant $C = C(m, s, q, M, n)$, with the following properties: For any function $f$ with d-dimensional input and one-dimensional output in subsets of the Sobolev space $W^{s,q}$:*

$$\|f\|_{W^{s,q}} \leq M,$$

*and for any $\epsilon \in (0, 1/2)$, there exists a ReLU MLP $\mathcal{N}$ such that:*

$$\|\mathcal{N} - f\|_{W^{n,q}} \leq \epsilon,$$

*and:*

$$
\begin{aligned}
\mathrm{size}(\mathcal{N}) &\leq C\epsilon^{-m/(s-n)} \log\left(\epsilon^{-s/(s-n)}\right), \\
\mathrm{depth}(\mathcal{N}) &\leq C \log\left(\epsilon^{-s/(s-n)}\right).
\end{aligned}
$$

Such error bounds can not be directly applied to the approximator $\mathcal{A}$ in our framework, since $\mathcal{A}$ is implemented by a single branch net $\boldsymbol{\beta}$ with $p$-dimensional output. It is different with stacking $p$ independent one-dimensional output networks $\{\mathcal{N}_j\}_{1 \leq j \leq p}$ and concatenating the outputs. The key difference lies in the parameter sharing of hidden layers. To fill the gap, we design a special structure of the branch net $\boldsymbol{\beta}$ without parameter sharing, as explained below.

Given $p$ independent one-dimensional output networks $\{\mathcal{N}_j : \mathbb{R}^m \to \mathbb{R}^1\}_{1 \leq j \leq p}$, denote the weight matrix of the $i$-th layer of the $\mathcal{N}_j$ as $\boldsymbol{W}_{i,j}$. The weight matrix of $i$-th layer of the branch net, $\boldsymbol{W}_i^{\boldsymbol{\beta}}$, can be constructed as:

$$
\boldsymbol{W}_1^{\boldsymbol{\beta}} = \begin{bmatrix} \boldsymbol{W}_{1,1} \\ \boldsymbol{W}_{1,2} \\ \vdots \\ \boldsymbol{W}_{1,p} \end{bmatrix}, \qquad \boldsymbol{W}_{i \geq 2}^{\boldsymbol{\beta}} = \begin{bmatrix} \boldsymbol{W}_{i,1} & 0 & \cdots & 0 \\ 0 & \boldsymbol{W}_{i,2} & \cdots & 0 \\ \vdots & \vdots & \ddots & \vdots \\ 0 & 0 & \cdots & \boldsymbol{W}_{i,p} \end{bmatrix}.
$$

The weight of first layer $\boldsymbol{W}_1^{\boldsymbol{\beta}}$ is a vertical concatenation of $\{\boldsymbol{W}_{1,j}\}_{1 \leq j \leq p}$. And the weight $\boldsymbol{W}_i^{\boldsymbol{\beta}}$ of any remaining layer $i \geq 2$ is a block diagonal matrix, with the main-diagonal blocks being $\{\boldsymbol{W}_{i,j}\}_{1 \leq j \leq p}$. It is easy to verify that such approximator is computationally equivalent to stacking of $p$ independent one-dimensional output networks $\mathrm{stack}(\{\mathcal{N}_j\}_{1 \leq j \leq p})$.

Let $q = 2$, $n = 0$, and $f = \mathcal{P} \circ \mathcal{G} \circ \mathcal{E}^{-1}$, then by Theorem 4 the approximator error is bounded by:

$$
\begin{aligned}
\widehat{\mathscr{E}}_{\mathcal{A}} &= \left\| \boldsymbol{\beta} - \mathcal{P} \circ \mathcal{G} \circ \mathcal{E}^{-1} \right\| \\
&= \left( \sum_j \left\| \mathcal{N}_j - \mathcal{P} \circ \mathcal{G} \circ \mathcal{E}_j^{-1} \right\|^2 \right)^{1/2} \\
&\leq (p\varepsilon^2)^{1/2} = \sqrt{p}\varepsilon.
\end{aligned}
$$

And the depth and size of $\boldsymbol{\beta}$ can be calculated by comparing with any $\mathcal{N}_j$:

$$
\mathrm{depth}(\mathcal{A}) = \mathrm{depth}(\mathcal{N}_j) \leq C \log\left(\varepsilon^{-1}\right),
$$

$$
\begin{aligned}
\mathrm{size}(\mathcal{A}) &= \mathrm{size}(\boldsymbol{W}_1^{\mathcal{A}}) + \sum_{i=2}^p \mathrm{size}(\boldsymbol{W}_i^{\mathcal{A}}) \\
&= p\,\mathrm{size}(\boldsymbol{W}^{1,j}) + \sum_{i=2}^p p^2\,\mathrm{size}(\boldsymbol{W}_{i,j}) \\
&\leq \sum_{i=1}^p p^2\,\mathrm{size}(\boldsymbol{W}_{i,j}) \\
&= p^2\,\mathrm{size}(\mathcal{N}_j) \\
&\leq Cp^2\varepsilon^{-m/s} \log\left(\varepsilon^{-1}\right).
\end{aligned}
$$

$\square$

Table 4: Typical control systems used in literature and our experiments and their dimensions.

| System | Description | Dynamics | $u$ # | $x$ # |
|---|---|---|---|---|
| Pendulum | single pendulum | ODE | 1 | 2 |
| RobotArm | two-link robotic arm | ODE | 1 | 4 |
| CartPole | one pendulum with cart | ODE | 1 | 4 |
| Quadrotor | helicopter with 4 rotors | ODE | 4 | 9 |
| Rocket | 6-DoF rocket | ODE | 3 | 9 |
| Heating | source heating on 2D plane | PDE | 1 | 1 |
| Pushing | pushing objects on various surfaces | ODE | 1 | 3 |
| StoPendulum | single pendulum with stochastic dynamics | SDE | 1 | 2 |

## D    EXTENSION TO NON-ZERO ENCODER ERROR

The approximation error estimation can be naturally extended to model non-zero encoder error, as derived in Lanthaler et al. (2022). For an encoder $\mathcal{E}$, its error $\widehat{\mathscr{E}}_{\mathcal{E}}$ is estimated by the distance to its optimal approximate inverse mapping, decoder $\mathcal{D}$, weighted by measure $\mu$.

$$\widehat{\mathscr{E}}_{\mathcal{E}} := \left( \int_F \| \mathcal{D} \circ \mathcal{E}(f) - f \|_F \mathrm{d}\mu(f) \right)^{\frac{1}{2}}$$

$$\mathcal{D} := \operatorname*{argmin}_{\mathcal{D}} \widehat{\mathscr{E}}_{\mathcal{E}}, \quad \text{s.t.} \quad \mathcal{E} \circ \mathcal{D} = \mathrm{Id}.$$

(13)

Similar to reconstructor error, this error quantifies the information loss during encoding. An ideal encoder should be invertible, i.e. the decoder $\mathcal{D} = \mathcal{E}^{-1}$, thus we have $\widehat{\mathscr{E}}_{\mathcal{E}} = 0$.

When the encoder error is non-zero, the definition of the approximator error $\widehat{\mathscr{E}}_{\mathcal{A}}$ (Eq. 8) should be modified accordingly, by replacing $\mathcal{E}^{-1}$ to $\mathcal{D}$:

$$\widehat{\mathscr{E}}_{\mathcal{A}} := \left( \int_{\mathbb{R}^m} \| \mathcal{A}(e) - \mathcal{P} \circ \mathcal{G} \circ \mathcal{D}(e) \|^2_{\ell^2(\mathbb{R}^p)} \mathrm{d}\left( \mathcal{E}_{\#}\mu \right)(e) \right)^{\frac{1}{2}}.$$

(14)

Also, the approximation error bound (Eq. 10) is changed to (proof similar to C.1, omitted here):

$$\widehat{\mathscr{E}} \leq C \left( \mathrm{Lip}(\mathcal{G}) \, \mathrm{Lip}(\mathcal{R} \circ \mathcal{P}) \widehat{\mathscr{E}}_{\mathcal{E}} + \mathrm{Lip}(\mathcal{R}) \widehat{\mathscr{E}}_{\mathcal{A}} + \widehat{\mathscr{E}}_{\mathcal{R}} \right).$$

(15)

## E    EXPERIMENT ENVIRONMENTS

### E.1    PENDULUM

$$\min_u \quad \int_0^{tf} \boldsymbol{c}_{\boldsymbol{x}}^{\top} (\boldsymbol{x}(t) - \boldsymbol{x}_{goal})^2 + c_u u^2(t) \, \mathrm{d}t$$

(16)

$$\text{s.t.} \quad \dot{\boldsymbol{x}}(t) = \begin{bmatrix} x_2(t) \\ [u(t) - m \cdot g \cdot l \sin x_1(t) - b \cdot x_2(t)]/I \end{bmatrix}, \quad \boldsymbol{x}(0) = \boldsymbol{x}_{init}$$

where state $\boldsymbol{x} = [x_1, x_2]^{\top}$, denoting the angle and angular velocity of the pendulum respectively, and control $u$ is the external torque. The initial condition $\boldsymbol{x}_{init} = [0, 0]^{\top}$, i.e. the pendulum starts from the lowest position with zero velocity. The cost functional consists of two parts: state mismatching penalty and control function regularization, and $\boldsymbol{c}_{\boldsymbol{x}} = [10, 1]^{\top}, c_u = 0.1$ are balancing coefficients. Other constants are: $m = 1, g = 10, l = 1, I = 1/3$.

### E.2    ROBOT ARM

$$\boldsymbol{M}(\boldsymbol{x}) = \begin{bmatrix} m_1 r_1^2 + I_1 + m_2(l_1^2 + r_2^2 + 2l_1 r_2 \cos(x_2)) & m_2(r_2^2 + l_1 r_2 \cos(x_2)) + I_2 \\ m_2(r_2^2 + l_1 r_2 \cos(x_2)) + I_2 & m_2 r_2^2 + I_2 \end{bmatrix},$$

$$\boldsymbol{C}(\boldsymbol{x}) = \begin{bmatrix} -m_2 l_1 r_2 \sin(x_2) x_4 & -m_2 l_1 r_2 \sin(x_2)(x_3 + x_4)) \\ m_2 l_1 r_2 \sin(x_2) x_3 & 0 \end{bmatrix},$$

$$\boldsymbol{g}(\boldsymbol{x}) = \begin{bmatrix} m_1 r_1 g \cos(x_1) + m_2 g(r_2 \cos(x_1 + x_2) + l_1 \cos(x_1)) \\ m_2 g r_2 \cos(x_1 + x_2) \end{bmatrix},$$

$$\begin{bmatrix} \dot{x}_1 \\ \dot{x}_2 \end{bmatrix} = \begin{bmatrix} x_3 \\ x_4 \end{bmatrix},$$

$$\begin{bmatrix} 0 \\ u \end{bmatrix} = \boldsymbol{M}(\boldsymbol{x}) \begin{bmatrix} \dot{x}_3 \\ \dot{x}_4 \end{bmatrix} + \boldsymbol{C}(\boldsymbol{x}) \begin{bmatrix} x_3 \\ x_4 \end{bmatrix} + \boldsymbol{g}(\boldsymbol{x}).$$

The RobotArm (also named Acrobot) is a planar two-link robotic arm in the vertical plane, with an actuator at the elbow. The state is $\boldsymbol{x} = [x_1, x_2, x_3 x_4]^\top$, where $x_1$ is the shoulder joint angle, and $x_2$ is the elbow (relative) joint angle, $x_3, x_4$ denotes their angular velocity respectively. The control $u$ is the torque at the elbow. Note that the last equation is the manipulator equation, where $\boldsymbol{M}$ is the inertia matrix, $\boldsymbol{C}$ captures Coriolis forces, and $\boldsymbol{g}$ is the gravity vector. The details of the derivation can be found in (Spong et al., 2020, Sec. 6.4).

The initial condition $\boldsymbol{x}_{init} = [\pi/4, \pi/2, 0, 0]^\top$, and the target state baseline $\boldsymbol{x}_{goal} = [\pi/2, 0, 0, 0]^\top$. The cost functional coefficients are $\boldsymbol{c_x} = [0.1, 0.1, 0.1, 0.1]^\top, c_u = 0.1$. Other constants are: mass of two links $m_{1,2} = 1$, gravitational acceleration $g = 0$, links length $l_{1,2} = 1$, distance from joint to the center of mass $r_{1,2} = 0.5$, moment of inertia $I_{1,2} = 1/3$.

### E.3 CART-POLE

$$\dot{x}_1 = x_3,$$
$$\dot{x}_2 = x_4,$$
$$\dot{x}_3 = \frac{1}{m_c + m_p \sin^2(x_2)} \left[ u + m_p \sin(x_2) \left( l x_4^2 + g \cos(x_2) \right) \right],$$
$$\dot{x}_4 = \frac{1}{l \left( m_c + m_p \sin^2(x_2) \right)} [-u \cos(x_2) - m_p l x_4^2 \cos(x_2) \sin(x_2) - (m_c + m_p) g \sin(x_2)].$$

In the CartPole system, an unactuated joint connects a pole(pendulum) to a cart that moves along a frictionless track. The pendulum is initially positioned upright on the cart, and the goal is to balance the pendulum by applying horizontal forces to the cart. The state is $\boldsymbol{x} = [x_1, x_2, x_3 x_4]^\top$, where $x_1$ is the horizontal position of the cart, $x_2$ is the counter-clockwise angle of the pendulum, $x_3$ velocity and angular velocity of cart and pendulum respectively. We refer the reader to of (Tedrake, 2022, Sec. 3.2) for derivation of above equations.

The initial condition $\boldsymbol{x}_{init} = [0, 0, 0, 0]^\top$, and the target state baseline $\boldsymbol{x}_{goal} = [0, \pi, 0, 0]^\top$. The cost functional coefficients are $\boldsymbol{c_x} = [0.1, 0.6, 0.1, 0.1]^\top, c_u = 0.3$. Other constants are: mass of cart and pole $m_{c,p} = 0.1$, gravitational acceleration $g = 10$, pole length $l = 1$.

### E.4 QUADROTOR

$$\dot{\boldsymbol{p}} = \boldsymbol{v},$$

$$m\dot{\boldsymbol{v}} = \begin{bmatrix} 0 \\ 0 \\ mg \end{bmatrix} + \boldsymbol{R}^\top(\boldsymbol{q}) \begin{bmatrix} 0 \\ 0 \\ \boldsymbol{1}^\top \boldsymbol{u} \end{bmatrix},$$

$$\dot{\boldsymbol{q}} = \frac{1}{2} \boldsymbol{\Omega}(\boldsymbol{\omega}) \boldsymbol{q},$$

$$\boldsymbol{J}\dot{\boldsymbol{\omega}} = \boldsymbol{T}\boldsymbol{u} - \boldsymbol{\omega} \times \boldsymbol{J}\boldsymbol{\omega}.$$

This system describes the dynamics of a helicopter with four rotors. The state $\boldsymbol{x} = [\boldsymbol{p}^\top, \boldsymbol{v}^\top, \boldsymbol{\omega}^\top]^\top \in \mathbb{R}^9$ consists of three parts: position $\boldsymbol{p}$, velocity $\boldsymbol{v}$, and angular velocity $\boldsymbol{\omega}$. The control $\boldsymbol{u} \in \mathbb{R}^4$ is the thrusts of the four rotating propellers of the quadrotor. $\boldsymbol{q} \in \mathbb{R}^4$ is the unit quaternion (Jia, 2019) representing the attitude(spacial rotation) of quadrotor w.r.t. the inertial frame. $\boldsymbol{J}$ is the moment of inertia in quadrotor's frame, and $\boldsymbol{T}\boldsymbol{u}$ is the torque applied to the quadrotor. Our setting is similar to (Jin et al., 2020, Appx. E.1), but we exclude the quaternion from the state.

The derivation is straightforward. The first two equations are Newton's laws of motion, the third equation is time-derivative of quaternion (Jia, 2019, Appx. B), and the last equation is Euler's rotation equation (Truesdell, 1992, Sec. I.10). And the coefficient matrices and operators used in the equations are defined as follows:

$$\boldsymbol{\Omega}(\boldsymbol{\omega}) = \begin{bmatrix} 0 & -\omega_1 & -\omega_2 & -\omega_3 \\ \omega_1 & 0 & \omega_3 & -\omega_2 \\ \omega_2 & -\omega_3 & 0 & \omega_1 \\ \omega_3 & \omega_2 & -\omega_1 & 0 \end{bmatrix},$$

$$\boldsymbol{R}(\boldsymbol{q}) = \begin{bmatrix} 1 - 2\left(q_3^2 + q_4^2\right) & 2\left(q_2 q_3 - q_4 q_1\right) & 2\left(q_2 q_4 + q_3 q_1\right) \\ 2\left(q_2 q_3 + q_4 q_1\right) & 1 - 2\left(q_2^2 + q_4^2\right) & 2\left(q_3 q_4 - q_2 q_1\right) \\ 2\left(q_2 q_4 - q_3 q_1\right) & 2\left(q_3 q_4 + q_2 q_1\right) & 1 - 2\left(q_2^2 + q_3^2\right) \end{bmatrix},$$

$$\boldsymbol{T} = \begin{bmatrix} 0 & -l/2 & 0 & l/2 \\ -l/2 & 0 & l/2 & 0 \\ c & -c & c & -c \end{bmatrix},$$

We set the initial state $\boldsymbol{x}_{init} = [[-8, -6, 9]^\top, \boldsymbol{0}, \boldsymbol{0}]^\top$, the initial quaternion $\boldsymbol{q}_{init} = \boldsymbol{0}$, and the target state baseline $\boldsymbol{x}_{goal} = \boldsymbol{0}$. Cost functional coefficients $c_{\boldsymbol{x}} = 1$, $c_u = 0.1$. Other constants are configured as: mass $m = 1$, wing length $l = 0.4$, moment of inertia $\boldsymbol{J} = 1$, z-axis torque constant $c = 0.01$.

### E.5 ROCKET

$$\dot{\boldsymbol{p}} = \boldsymbol{v},$$

$$m\dot{\boldsymbol{v}} = \begin{bmatrix} mg \\ 0 \\ 0 \end{bmatrix} + \boldsymbol{R}^\top(\boldsymbol{q})\boldsymbol{u},$$

$$\dot{\boldsymbol{q}} = \frac{1}{2}\boldsymbol{\Omega}(\boldsymbol{\omega})\boldsymbol{q},$$

$$\boldsymbol{J}\dot{\boldsymbol{\omega}} = \boldsymbol{T}\boldsymbol{u} - \boldsymbol{\omega} \times \boldsymbol{J}\boldsymbol{\omega}.$$

The rocket system models a 6-DoF rocket in 3D space. The formulation is very close to Quadrotor mentioned above. The state $\boldsymbol{x} = [\boldsymbol{p}^\top, \boldsymbol{v}^\top, \boldsymbol{\omega}^\top]^\top \in \mathbb{R}^9$ is same as that of Quadrotor, but the control $\boldsymbol{u} \in \mathbb{R}^3$ is slightly different. Here $\boldsymbol{u}$ denotes the total thrust in 3 dimensions. Accordingly, the torque $\boldsymbol{T}\boldsymbol{u}$ is changed to:

$$\boldsymbol{T}\boldsymbol{u} = \begin{bmatrix} -l/2 \\ 0 \\ 0 \end{bmatrix} \times \boldsymbol{u}$$

We set the initial state $\boldsymbol{x}_{init} = [10, -8, 5, -1, \boldsymbol{0}]^\top$, the initial quatenion $\boldsymbol{q}_{init} = [\cos(0.75), 0, 0, \sin(0.75)]^\top$, and the target state baseline $\boldsymbol{x}_{goal} = \boldsymbol{0}$. The cost functional coefficients $c_{\boldsymbol{x}} = 1$, $c_u = 0.4$. Other constants are configured as: mass $m = 1$, rocket length $l = 1$, the moment of inertia $\boldsymbol{J} = \text{diag}([0.5, 1, 1])$

### E.6 HEATING

$$-\Delta x = u \quad \text{in } \Omega$$
$$x = 0 \quad \text{on } \partial\Omega$$

The heating system mimics a 2D plane $\Omega$, whose temperature $x$ is controlled by a heating source $u$, such as a plane heated by electromagnetic induction or microwaves (Tröltzsch, 2010, Chap. 1.2.1). The dynamics is a Poisson equation with zero boundary, where $\Delta = \nabla \cdot \nabla = \nabla^2$ is the Laplace operator. The objective is thus a double integral over $\Omega$:

$$\min_u \int_\Omega c_x(x(\boldsymbol{s}) - x_{goal}(\boldsymbol{s}))^2 + c_u u^2(\boldsymbol{s})\, \mathrm{d}\boldsymbol{s}\,.$$

The target state here is a function, not a vector. To simplify the problem, we let $x_{goal}(\boldsymbol{s}) = a\sin(\pi s_1)\sin(\pi s_2) + b\sin(\pi s_1) + c\sin(\pi s_2)$, with $[a, b, c]^\top$ being the parameters. Let the baseline parameters to be $[a, b, c]^\top = \boldsymbol{1}$, and the in and out of distribution problem are generated by adding noise $\boldsymbol{\epsilon}_{in}$ and $\boldsymbol{\epsilon}_{out}$ to the baseline. The cost functional coefficients are $c_x = 1, c_u = 10^{-4}$. During testing, the area $\Omega = [0, 1]^2$ is evenly divided into 100 areas, with 121 grid points.

| Shape |  |
|---|---|
| | `rect1, rect2, rect3, hex, ellip1, ellip2, ellip3 , butter , tri1, tri2, tri3` |
| Surface | `abs, derlin, polywood, pu` |
| Speed (mm/s) | 10, 20, 50, 75, 100, 150, 200, 300, 400, 500 |
| Acceleration ($\text{ms}^{-2}$) | 0, 0.1, 0.2, 0.5, 0.75, 1, 1.5, 2, 2.5 |
| Initial contact | 33 points for `tri1-3` and `hex`, 40 for `ellip1-3` and `butter`, and 44 for `rect1-3` |
| Initial push direction | 0°, 20°, 40°, 60°, 80°, -20°, -40°, -60°, -80° |

Figure 6: List of variables explored in Pushing dataset, credited to Yu et al. (2016).

### E.7 STOCHASTIC PENDULUM

$$\mathrm{d}x_1(t) = x_2(t)\,\mathrm{d}t\,,$$
$$\mathrm{d}x_2(t) = \frac{1}{I}[u(\boldsymbol{o}(t)) - mgl\sin x_1(t)]\,\mathrm{d}t + \sigma\mathrm{d}B(t),$$
$$\boldsymbol{o}(t) = \begin{bmatrix} \sin x_1(t) \\ \cos x_1(t) \\ x_2(t) \end{bmatrix}.$$

This system is similar to the simple Pendulum system introduced in the main text, but with Brownian motion $B(t)$ (Uhlenbeck & Ornstein, 1930) involved in the dynamics, resulting in a stochastic OCP (Fleming & Rishel, 2012). The state function is now a random process, thus the cost functional is defined as an expectation:

$$\min_u \quad E \int_0^{tf} \boldsymbol{c}_{\boldsymbol{x}}^\top (\boldsymbol{x}(t) - \boldsymbol{x}_{goal})^2 + c_u u^2(\boldsymbol{o}(t))\,\mathrm{d}t\,.$$

The state is defined by angle $x_1$ and angular velocity $x_2$, and control $u$ is the torque applied to the pendulum. We follow the convention of Gym environment Brockman et al. (2016) by adding $\boldsymbol{o}$, the observation of state $\boldsymbol{x}$. And further constrain the scale of state and control as $|u| \leq 2, |x_2| \leq 8$. The initial state is $\boldsymbol{x}_{init} = \boldsymbol{0}$, and the target state baseline is $\boldsymbol{x}_{goal} = [\pi, 0]^\top$. Cost functional coefficients $\boldsymbol{c}_{\boldsymbol{x}} = \boldsymbol{1}$, $c_u = 0.001$. Other constants are: mass $m = 1$, length $l = 1$, gravitational acceleration $g = 10$, scale of noise $\sigma = 0.01$.

We solve the problem in a closed-loop optimal control scheme, where the model takes the current state from the environment as input at each time index, then outputs the control to the environment. This setting is the same as RL, if we define the reward of RL as the negative cost of OCP.

### E.8 PUSHING

In this dataset, the robot executes an open-loop straight push along a straight line of 5 cm, with different shapes of objects, materials of surface, velocity and accelerations, and contact positions and angles, see Fig. 6 for details.

The control and state trajectories are recorded at 250 Hz. The length of the recorded time horizon varies among samples, due to the difference in velocity and acceleration. We select acceleration $a = 0.5ms^{-2}$, with initial velocity $v = 0$. Then define time horizon $T = 0.44s$, and extract 110 time indices per instance.

The input to the encoder is 4167-dim, including a 768-dim gray-scale image of the shape, 3280-dim friction map matrix, 110-dim trajectory, and 9-dim other parameters (e.g. mass and moment of inertia). The input to the neural network (including CNN) is 801-dim, and the encoded vector $\boldsymbol{e}$ is 44-dim.

## F  DETAILS ON IMPLEMENTATIONS

For all the systems, we have 2,000 samples in each ID/OOD validation set, and 100 problems in each ID/OOD benchmark set. The size of the training set varies among systems, and is roughly proportional to the number of trainable parameters, as displayed in Table 5.

Table 5: Hyper-parameter settings of the proposed OptCtrlOP for different systems.

| System | Depth | #Params | #Train Data | #Epochs |
|---|---|---|---|---|
| Pendulum | 7 | 7681 | $1 \times 10^4$ | $1 \times 10^4$ |
| RobotArm | 2 | 3601 | $1 \times 10^4$ | $1 \times 10^4$ |
| CartPole | 3 | 6881 | $3 \times 10^4$ | $2 \times 10^3$ |
| Quadrotor | 3 | 17284 | $1 \times 10^5$ | $5 \times 10^2$ |
| Rocket | 3 | 13883 | $1 \times 10^5$ | $5 \times 10^2$ |
| Heating | 2 | 3601 | $1 \times 10^4$ | $1 \times 10^4$ |
| Pushing | 3 | 12202 | $5 \times 10^4$ | $2 \times 10^3$ |
| StoPendulum | 3 | 6881 | $1 \times 10^5$ | $1 \times 10^4$ |

PDP, DM, and synthetic control systems are implemented in CasADi (Andersson et al., 2019), which are adapted from the code repository[1]. For classical methods, the dynamics are discretized by Euler method. To limit the running time, we set the maximum number of iterations of PDP to 2,500.

OptCtrlOP and other neural models are implemented in PyTorch (Paszke et al., 2019).

For MLP, hyper-parameters are the same or as close as to that of OptCtrlOP. We adjust the width of MLP layer to reach almost the same number of parameters. For FNO[2], we set the number of Fourier layers to 4 as suggested in the open-source codes, and tune the network width such that the number of parameters is in the same order as that of OptCtrlOP. Notice that the original FNO outputs function values at fixed time indices, which is inconsistent with our experiment setting. Thus we slightly modify it by adding time indices to its input. For GEN[3], we set 9 graph nodes uniformly spaced in time (or space) horizon, and perform 3 graph convolution steps on them. The input function initializes the node features at time index $t = 0$ (multiplied by weights). For any time index $t$, the GEN output is defined as the weighted average of all node features. Both input/output weights are softmax of negative distances between $t$ and node positions.

For stochastic OCP (i.e. Stochastic Pendulum), we choose a reinforcement learning algorithm named Proximal Policy Optimization(PPO) (Schulman et al., 2017) as the ground truth closed-loop OCP solver, by defining the reward as a negative cost. The PPO implementation is credited to open-source package *Stable Baselines* (Raffin et al., 2021), with hyper-parameter settings cited from Raffin (2020).

For fairness, all training/testing cases are executed on an Intel i9-10920X CPU, without GPU.

# G MORE EXPERIMENT RESULTS

Table 6: Results of Pendulum environment.

| | Time (sec./instance) | ID MAPE | OOD MAPE | ID-OOD Gap |
|---|---|---|---|---|
| DM | $3.80 \times 10^{-2}$ | \ | \ | \ |
| Ours | $3.33 \times 10^{-7}$ | $3.05 \times 10^{-4}$ | $4.09 \times 10^{-3}$ | $3.79 \times 10^{-3}$ |
| MLP | $3.40 \times 10^{-5}$ | $4.96 \times 10^{-4}$ | $1.67 \times 10^{-2}$ | $1.62 \times 10^{-2}$ |
| GEN | $6.19 \times 10^{-5}$ | $5.61 \times 10^{-3}$ | $4.46 \times 10^{-2}$ | $3.90 \times 10^{-2}$ |
| FNO | $3.47 \times 10^{-4}$ | $3.62 \times 10^{-4}$ | $3.28 \times 10^{-3}$ | $2.91 \times 10^{-3}$ |
| PDP | $5.29 \times 10^1$ | $2.79 \times 10^{-1}$ | $1.04 \times 10^{-1}$ | $1.75 \times 10^{-1}$ |

---

[1] https://github.com/wanxinjin/Pontryagin-Differentiable-Programming/tree/master
[2] https://github.com/zongyi-li/fourier_neural_operator
[3] https://github.com/FerranAlet/graph_element_networks

Table 7: Results of RobotArm environment.

|      | Time (sec./instance)     | ID MAPE               | OOD MAPE              | ID-OOD Gap            |
|------|--------------------------|-----------------------|-----------------------|-----------------------|
| DM   | $4.58 \times 10^{-2}$    | ╲                     | ╲                     | ╲                     |
| Ours | $1.88 \times 10^{-7}$    | $1.24 \times 10^{-5}$ | $5.95 \times 10^{-5}$ | $4.70 \times 10^{-5}$ |
| MLP  | $1.56 \times 10^{-5}$    | $1.16 \times 10^{-5}$ | $1.43 \times 10^{-3}$ | $1.42 \times 10^{-3}$ |
| GEN  | $6.24 \times 10^{-5}$    | $1.53 \times 10^{-5}$ | $1.69 \times 10^{-4}$ | $1.53 \times 10^{-4}$ |
| FNO  | $3.48 \times 10^{-4}$    | $8.86 \times 10^{-6}$ | $1.29 \times 10^{-4}$ | $1.20 \times 10^{-4}$ |
| PDP  | $5.62 \times 10^{1}$     | $4.73 \times 10^{-3}$ | $1.43 \times 10^{-2}$ | $9.55 \times 10^{-3}$ |

Table 8: Results of CartPole environment.

|      | Time (sec./instance)     | ID MAPE               | OOD MAPE              | ID-OOD Gap            |
|------|--------------------------|-----------------------|-----------------------|-----------------------|
| DM   | $4.63 \times 10^{-2}$    | ╲                     | ╲                     | ╲                     |
| Ours | $1.99 \times 10^{-7}$    | $1.60 \times 10^{-5}$ | $4.79 \times 10^{-5}$ | $3.19 \times 10^{-5}$ |
| MLP  | $2.02 \times 10^{-5}$    | $3.79 \times 10^{-5}$ | $8.97 \times 10^{-5}$ | $5.18 \times 10^{-5}$ |
| GEN  | $6.41 \times 10^{-5}$    | $2.35 \times 10^{-5}$ | $9.09 \times 10^{-4}$ | $8.85 \times 10^{-4}$ |
| FNO  | $3.49 \times 10^{-4}$    | $1.82 \times 10^{-5}$ | $4.98 \times 10^{-5}$ | $3.16 \times 10^{-5}$ |
| PDP  | $5.61 \times 10^{1}$     | $5.96 \times 10^{-4}$ | $4.52 \times 10^{-5}$ | $5.50 \times 10^{-4}$ |

Table 9: Results of Rocket environment.

|      | Time (sec./instance)     | ID MAPE               | OOD MAPE              | ID-OOD Gap            |
|------|--------------------------|-----------------------|-----------------------|-----------------------|
| DM   | $9.71 \times 10^{-2}$    | ╲                     | ╲                     | ╲                     |
| Ours | $3.37 \times 10^{-7}$    | $4.42 \times 10^{-5}$ | $5.24 \times 10^{-4}$ | $4.80 \times 10^{-4}$ |
| MLP  | $2.87 \times 10^{-5}$    | $5.71 \times 10^{-5}$ | $5.83 \times 10^{-4}$ | $5.26 \times 10^{-4}$ |
| GEN  | $6.16 \times 10^{-5}$    | $3.57 \times 10^{-5}$ | $6.00 \times 10^{-4}$ | $5.65 \times 10^{-4}$ |
| FNO  | $6.33 \times 10^{-4}$    | $7.03 \times 10^{-5}$ | $1.81 \times 10^{-4}$ | $1.11 \times 10^{-4}$ |
| PDP  | $7.01 \times 10^{1}$     | $1.80 \times 10^{-7}$ | $1.52 \times 10^{-7}$ | $2.79 \times 10^{-8}$ |

Table 10: Results of Heating environment.

|      | Time(s/problem)          | ID MAPE               | OOD MAPE              | ID-OOD Gap            |
|------|--------------------------|-----------------------|-----------------------|-----------------------|
| DM   | $5.87 \times 10^{-2}$    | ╲                     | ╲                     | ╲                     |
| Ours | $2.14 \times 10^{-7}$    | $1.07 \times 10^{-2}$ | $1.14 \times 10^{-2}$ | $7.00 \times 10^{-4}$ |
| MLP  | $2.05 \times 10^{-5}$    | $1.93 \times 10^{-2}$ | $1.67 \times 10^{-2}$ | $2.60 \times 10^{-3}$ |
| GEN  | $6.70 \times 10^{-5}$    | $1.67 \times 10^{-1}$ | $2.20 \times 10^{-1}$ | $5.30 \times 10^{-2}$ |
| FNO  | $4.37 \times 10^{-4}$    | $2.43 \times 10^{-2}$ | $1.00 \times 10^{-2}$ | $1.43 \times 10^{-2}$ |

Table 11: Results of closed-loop control on StochasticPendulum environment. The ground truth result is generated by PPO.

|          | Time (sec./instance)                          | In Dist. MAPE                                  | Out of Dist. MAPE                              |
|----------|-----------------------------------------------|------------------------------------------------|------------------------------------------------|
| PPO      | $3.81 \times 10^{1}(\pm 1.86 \times 10^{-1})$ | ╲                                              | ╲                                              |
| MLP      | $2.10 \times 10^{-4}(\pm 1.87 \times 10^{-6})$ | $9.13 \times 10^{-2}(\pm 4.59 \times 10^{-2})$ | $4.69 \times 10^{-2}(\pm 1.07 \times 10^{-2})$ |
| OptCtrlOP | $3.33 \times 10^{-4}(\pm 4.08 \times 10^{-6})$ | $9.19 \times 10^{-2}(\pm 4.59 \times 10^{-2})$ | $3.04 \times 10^{-2}(\pm 1.09 \times 10^{-2})$ |

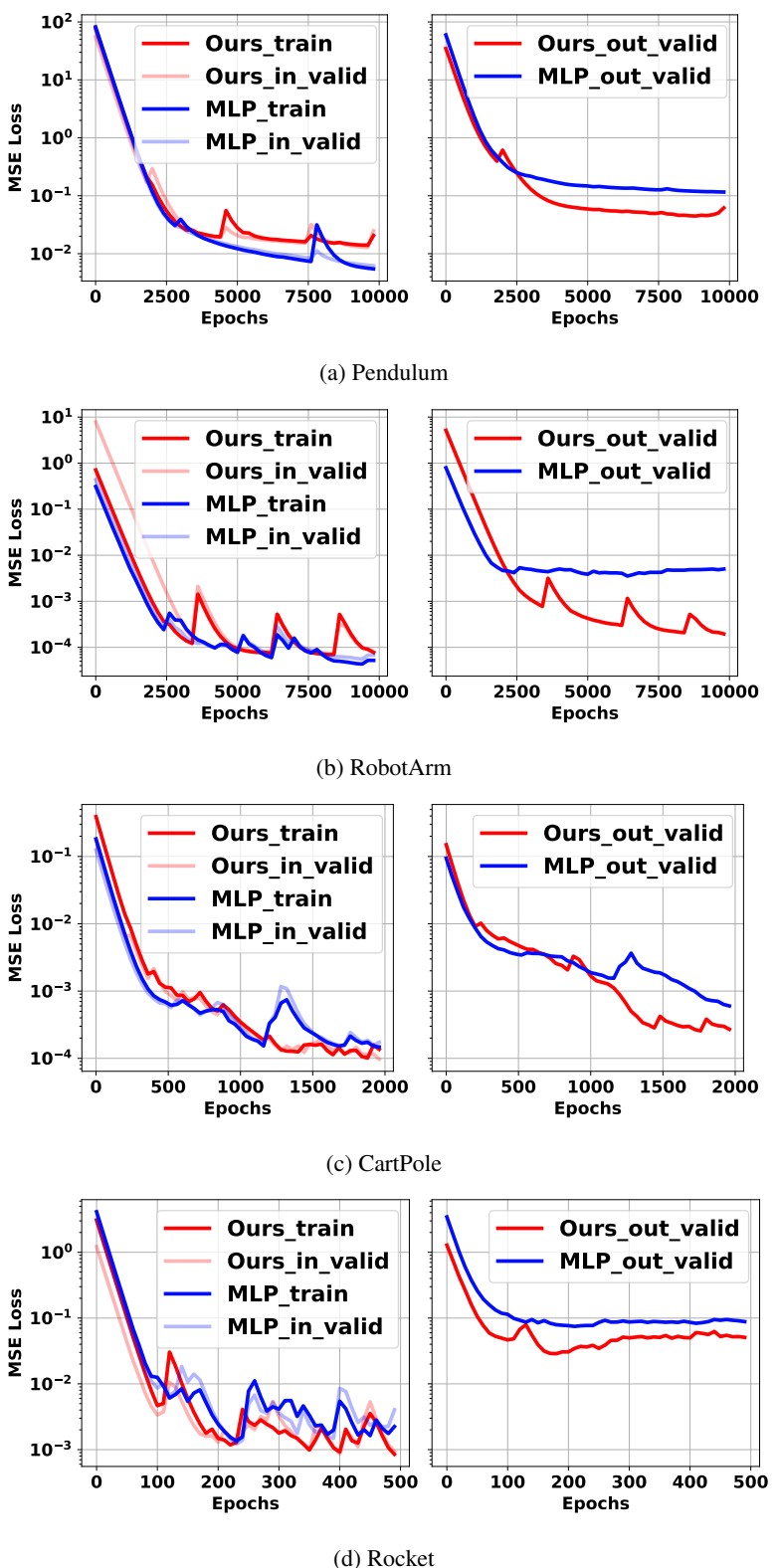

Figure 7: The loss curves of 4 systems on the training set, in- and out-of-distribution testing sets. All curves are visualized after exponential moving average with weight=0.5

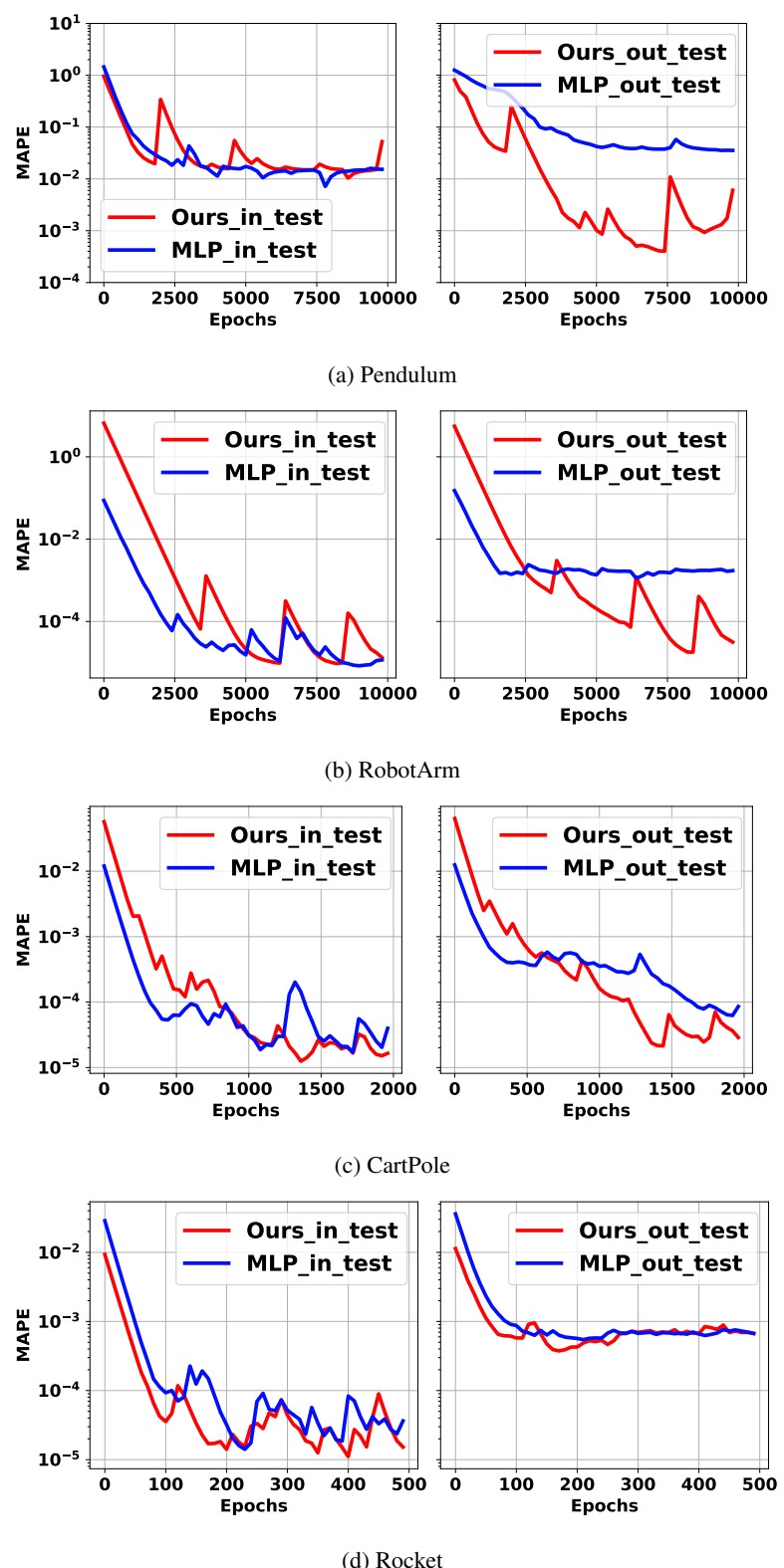

(a) Pendulum

(b) RobotArm

(c) CartPole

(d) Rocket

Figure 8: The cost MAPE (mean absolute percentage error) curves of 4 systems on the in- and out-of-distribution testing sets. All curves are visualized after exponential moving average with weight=0.5

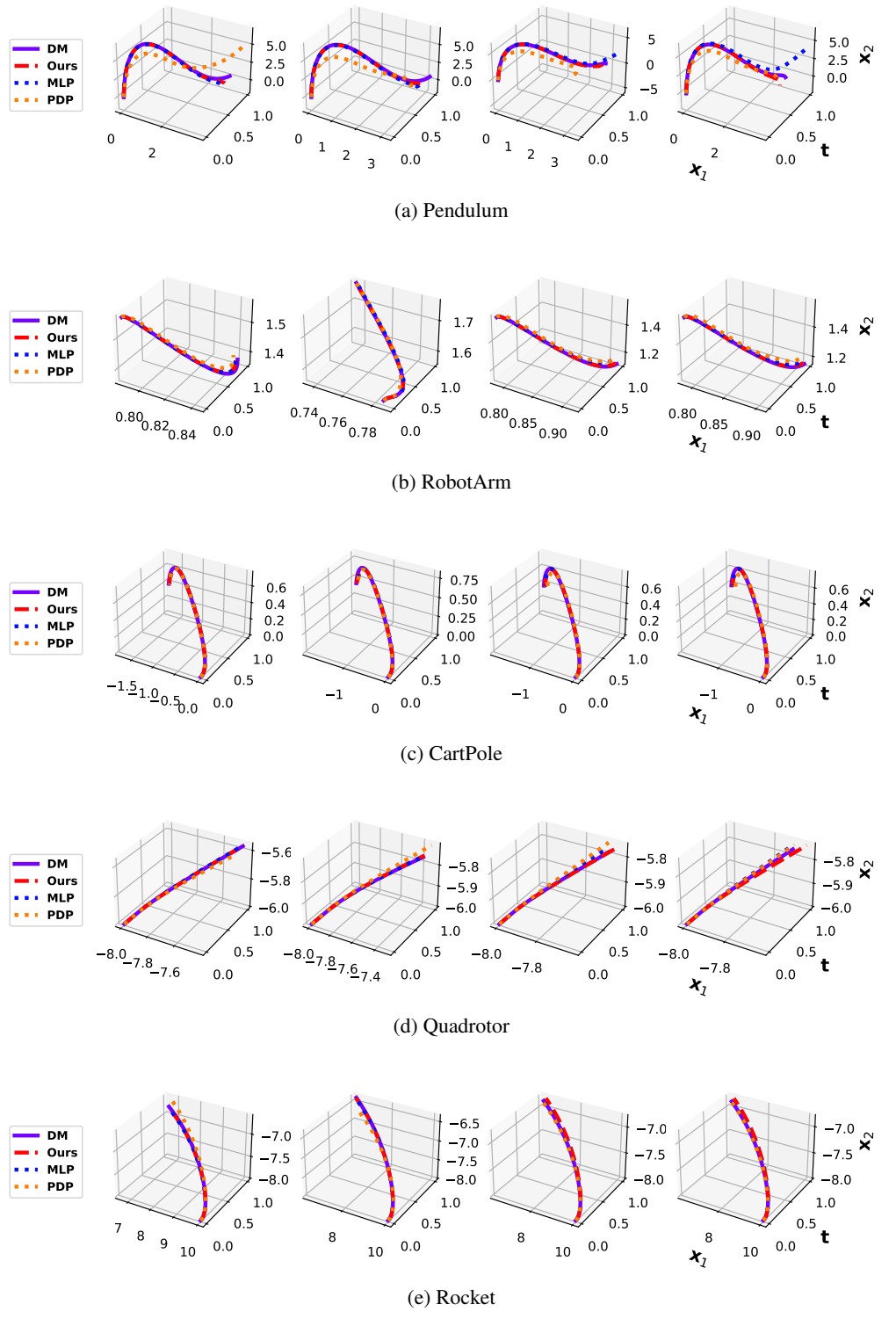

Figure 9: The state trajectories of five systems on four randomly sampled OCPs. The left two columns are in distribution problems, and the right two columns are out of distribution problems. The three dimensions consist of time $t$ and the first two dimensions of state $x_1, x_2$.

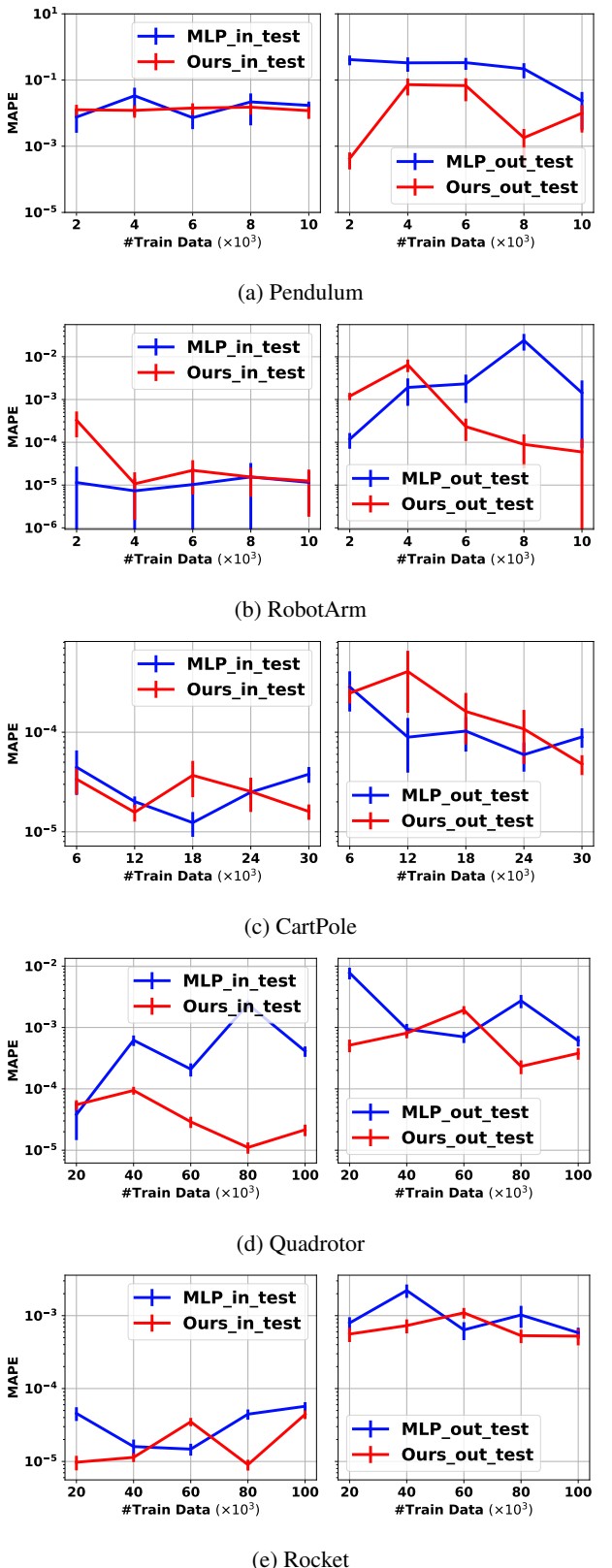

Figure 10: The MAPE (mean absolute percentage error) w.r.t. the number of training samples curves of 5 systems on the in- and out-of-distribution testing sets.

