# OpenReview forum: "Learning Instance-Solution Operator For Optimal Control"
_ICLR.cc/2023/Conference — Submitted to ICLR 2023_

### Official Review · Reviewer_p9mD · 2022-10-24

**Confidence:** 4
**Clarity, Quality, Novelty And Reproducibility:** See above
**Correctness:** 2
**Technical Novelty And Significance:** 3
**Empirical Novelty And Significance:** 3
**Recommendation:** 5

**Strength And Weaknesses:**

Notes on Each section:

Related Work and Baseline:

Weakness:  When talking about neural operators, only the DeepONet is mentioned but another related line of work Fourier Neural Operator (https://arxiv.org/pdf/2010.08895.pdf) is not mentioned, nor compared in the experiment section. In addition, the authors mentioned about another neural operator work Graph neural operator https://arxiv.org/pdf/2003.03485.pdf, still, no comparison of the GNO to DeepONet is conducted when learning the optimal control operators. The authors mentioned GNO may has high computational complexity this slower computational time, but a comparison between DeepONet, FNO, and GNO in learning the optimal control operator is needed to justify the choice of network structure.

Methodology:

Strength: The methodology section is well explained and easy to follow.

Weakness:

* One of the major weaknesses of this paper is that the approximation error bound in Section 3 does not match the experiment setting in Section 4. The author mentioned that “if the encoder is implemented by functional approximations e.g. point-wise evaluation, then the error should be considered”. In fact, their experiments do use a discretization grid/point-wise evaluation, so the approximation bound in Section 3 should match the actual setting and include the error from the encoder.

* V needs to be defined in Eqn. 8

Experiments:

Weakness:

* My biggest concern is regarding the comparison to necessary baselines in order to benchmark the proposed approach. The authors only compared the DeepONet against MLP. They need to compare against other neural operators including both FNO and Graph-based Neural Operator, in terms of both accuracy and computation speed.

* Related to the first weakness point, another useful baseline to compare against is PINN. Though the authors mentioned in related work that PINNs need to solve instance by instance, while DeepONet learns the entire operator. However, how about learning accuracy? Similarly, as my first comment, such a comparison between the proposed method and PINN in terms of both accuracy and computational efficiency is necessary to justify the choice of network structure. Especially on data that is unseen during the training stage or completely Out of Distribution test data, PINN might have better accuracy than neural operators since it solves the equation rather than suffering from generalization error. I would love to see such a comparison on test data that are sufficiently different from the training case and see the comparison between the proposed approach and PINN.

* How can we tell which method is better from the state trajectories plot (e.g., Fig. 4 and Fig. 7)? Why is one trajectory better than another?

* I don’t understand how the data for the stochastic pendulum was created using RL. “For stochastic OCP (i.e. Stochastic Pendulum), we choose a reinforcement learning algorithm named Proximal Policy Optimization(PPO) (Schulman et al., 2017) as the ground truth OCP solver. ” Page 21. The authors need to provide sufficient descriptions and details if they want to include this part of the results.

* Another thing was I think the authors ran out of space. There is no extensive discussion about the real-world examples provided.

Small questions/weaknesses:
* Why vary the time horizon? What did this achieve in demonstrating?
* Why are the noise distributions for the pendulum different? One is positive and the other is negative (epsilon_in and epsilon_out)
* Why is the batch size 10k for a data set of 1k (Pendulum example) - which seems like a typo



**Summary Of The Paper:**

In this paper, the authors propose a new application of neural operators. They target a well-studied problem of optimal control where there is a dynamical system and the goal is to find a control function that minimizes a certain cost specified by the control system.

To do this, they first transform the OCP (optimal control problem) into a BVP by utilizing a known technique in optimal control known as the Pontryagins Maximum Principle. Then they apply a DeepONet-based operator on the resulting BVP. The authors then use previous work on DeepONet approximation bounds to provide a new theorem on the approximation error of their approach. They then provide simulations
of multiple systems as their set of experiments (mainly focused on a pendulum).

**Summary Of The Review:**

This paper proposes a new application of neural operators in learning the optimal deterministic control operator derived from the Pontryagins Maximum Principle. The problem of the study is interesting. The main limitation/weakness of the proposed paper is as follows:

1) the theory analysis in Section 3 does not match the actual method. The authors should include the encoder approximation error to the approximation error bound;

2) Lack of necessary comparison to baselines to justify the choice of the model architecture. The authors need to compare against FNO, GNO, and PINN in terms of computational efficiency as well as approximation accuracy (in both in-distribution data and OOD samples) to benchmark the proposed approach and argue why such a design is desired;

3) Except for the pendulum example, other experiments are not discussed in necessary detail - the authors may want to re-assess the distribution of space to include at least one more real-world like control example and comparison in the main text beside the pendulum example.

---

> ### Author Response · Authors · 2022-11-20
> **In response to Reviewer p9mD**
>
> We especially thank you for your careful reading. We are grateful that you proposed so many constructive suggestions. The proposed comments are informative and valuable. All your suggestions will help us further improve the quality of this paper. We will address your concerns point-by-point.
>
> > ***Q1**: The approximation error bound(ignoring encoder error) does not match the experiment setting (using point-wise evaluation).*
>
> **Re-1**: All synthetic environments of the original submission do not involve point-wise evaluation. The target state function is a constant function (i.e. a vector), and the encoder directly outputs the vector, thus the encoder error is zero. In the newly added real-world pushing dataset (Section 3.2), we use a more complicated encoder, including a smoothing filter (along with down-sampling) and a CNN. The encoder error now is non-zeros but the analysis framework of approximation error can be extended to include this error (Appx. D). The estimation of encoder error itself depends on the specific choice of the encoder, which is beyond the scope of this paper and is left to future work.
>
>
> > ***Q2**: Necessary neural baselines: FNO/GNO/PINN.*
>
> **Re-2**: We have added FNO and GEN (the primary version of GNO) as baselines. However, it should be noticed that neural operators like FNO and GEN are not designed for learning control operators, but for learning PDEs. The input of FNO and GEN should be the point-wise evaluation of function, but the input of the control operator is the encoding of cost functional and dynamics (e.g. the target state, the encoding of object shape. etc). The backbone architecture, i.e. Fourier networks and GNN, can work as a substitution for the DeepONet backbone, under our optimal control operator perspective. However, the resulting models lose their physical priors. For example, the original FNO performs Fourier transform on the input function, and learns the parametric mapping in the frequency space. However, for an encoding vector(e.g. in the pushing dataset, the encoding of object shape, friction and trajectory, and other parameters ), the Fourier transform does not have a physical interpretation, and the frequency space is also undefined. Thus the Fourier layers degenerate to general 1-d convolution layers, and the FNO degenerates into a general neural operator consisting of MLP and 1d-CNN.
>
> The GEN also faces the same problem as FNO. In the original GEN, the graph nodes are defined as grids on a time or space horizon, and the node features are initialized by the evaluation of the input function at an arbitrary grid position. Such a scheme can not naturally handle an OCP encoding vector that is not generated by grid-wise evaluation. In our implementation, we set a dummy grid position for the input. Thus the GEN degenerates into a general neural operator consisting of MLP.
>
> When it comes to the PINN, we have implemented an MLP-based PINN and a DeepONet-based PINN, and tested their performance on the Pendulum. In theory, PINNs should have better accuracy on ID/OOD, since they solve OCP instance by instance. However, it turns out that both versions of PINN fail to output reasonable results. When it converges, the PINN loss is greater than $1\times10^3$, and the MAPE is about 1, while a reasonable MAPE should be lower than $10^{-3}$. The failure may result from the imbalanced loss terms, as mentioned in the introduction of PINN (Appx. B.2).
>
>
> > ***Q3**: How can we tell which method is better from the state trajectories plot (Fig. 4 and 7 in the original submission)?*
>
> **Re-3**: We leave the trajectory plots in Fig.9 in the appendix. We decide to treat those plots as merely a demonstration, instead of a performance comparison, since the number of samples is very limited.
>
> > ***Q4**:how the data for the stochastic pendulum created using RL?*
>
> **Re-4**: For the stochastic pendulum, the OCP is solved in the closed-loop (feedback) control scheme. At each time stamp $t_i$, the model takes current state $x(t_i)$ as input, outputs the control $u(t_i)$ to the environment, and continues to the next time stamp. This setting is the same as RL if we set the negative of the total cost as the reward. We have added more explanation in Appx. E.7 and Appx. F.

---

> ### Author Response · Authors · 2022-11-20
> **In response to Reviewer p9mD (continued)**
>
> > ***Q5**: need more real-world control examples.*
>
> **Re-5**: We switch the Pendulum experiment for the Quadrotor, whose dynamic functions are more complicated. In addition, we perform an experiment on a real-world object-pushing dataset, to evaluate the generalization ability of OptCtrlOP. The dataset records the forces and motion of a controlled robotic arm pushing objects on various surfaces. In our experiment, the cost functional, dynamics, and initial state are all variable, and the ID/OOD is defined on the initial state. The dynamics of the pushing problem are hard to model since the friction of the surface is not uniform. In such a complex task without explicit expression, the classical direct or indirect methods, such as DM and PMP, are not applicable. However, OptCtrlOP is able to learn from demonstration data, providing relatively accurate predictions compared with all neural baselines.
>
> > ***Q6**: minor questions*
> > 1) an undefined v in Eq. 8
> > 2) Why vary the time horizon?
> > 3) Why are the noise distributions $\epsilon_{in}, \epsilon_{out}$ different?
> >  4) Why is the batch size 10k for a data set of 1k (Pendulum)?
>
> **Re-6**:
> 1. We fixed the typo on Eq. 8
> 2. The varying time horizon is a small implementation trick. If the time horizon is fixed, then the time indices of all samples are also fixed to a finite set of points, since the time indices of each sample are defined as uniform grids of the time horizon. But if we slightly perturb the time horizon, then the time indices of all samples are freely spaced in the whole horizon.
> 3. the noise distributions are different to distinguish in-distribution samples and out-of-distribution (ID and OOD) datasets.
> 4. The sizes of training datasets of all experiments are greater than or equal to 10k, see Tab. 5 in the appendix.

---

### Official Review · Reviewer_Yqn8 · 2022-11-04

**Confidence:** 3
**Correctness:** 3
**Technical Novelty And Significance:** 2
**Empirical Novelty And Significance:** Not applicable
**Recommendation:** 5

**Clarity, Quality, Novelty And Reproducibility:**

**Clarity/Quality**: The paper was well organized and written. The overall production quality of the paper is good and easy to follow. I believe most of the paper is also clear. I mentioned some sections in the Strengths and Weaknesses section that would benefit from more explanation. In particular, if there are any assumptions or claims about the approach (e.g. related to the initial conditions previously discussed) I recommend stating/citing those.

**Novelty**: The work expands upon previous work in the field of using networks to approximate operators. The primary contribution appears to be a related network architecture applied to optimal control problems. With my current understanding of the initial condition limitation, I do not believe this is a significant contribution. The method, as I understand it, uses a neural network architecture in an efficient way to estimate optimal controls from a single initial state. The network required a set of optimal controls to train on and the optimal controls were generated from another method that could perform real-time replanning at different states (e.g. SCP DM).

**Reproducibility**: The paper provides many details to be able to recreate the work. Additionally, the authors that the code will be made available. I did not try to reproduce the results, but I currently do not have any concerns about reproducibility.


**Strength And Weaknesses:**

The paper does a good job of providing background information and providing a theoretical analysis of their approach. I also appreciate how the paper is organized and includes both theoretical and empirical results. However, I think the largest weakness is the lack of evidence to support the strong claims of the paper.

- Claim: "[OptCtrlOP] solves OCPs in a one-shot manner with no dependence on the explicit expression of dynamics or iterative optimization processes."

  - While I understand how this claim can be reached based on the wording, I believe it is a bit misleading. As presented, the method requires quite a bit of data involving the optimal control of the system. While the execution of the operator is not depending on the expression of the dynamics or an iterative nature, developing the mapping and training certainly do.

- Claim: "The design is in principle endowed with substantial speedup in running time..."

  - Results from the simulations support this claim, but I do not believe the baselines support such a claim. The speed shown for inference of OptCtrlOP is impressive and I believe would still be the better performer. However, the performance shown for the direct method appears to be slow compared to existing methods in that category. I understand the use of IPOPT as a generic non-linear solver; however, most implementations of a DM involve more specific implementations. I recommend a comparison of the systems to SCP-based implementations closer aligned with current state-of-the-art implementations. Additionally, do you expect this performance disparity to scale as the problem complexity increases? In Figure 2, there it appears there is more than a 2x increase in run time when going from problems with an input size of one to four (pendulum/robot arm/cart pole to quadrotor). This leads me to wonder how the inference time scales as the network size needs to grow.

- Claim: "... the model reusability is guaranteed by high-quality in- and out-of-distribution generalization."
  - The discussion about model reusability was confusing for me to follow and see the connection to this claim. The experiments did provide a look at what happens with the goal location was outside of the training distribution. However, what about other model parameters (e.g. length of the pendulum, weight, other physical parameters)? Would you expect the same performance with modifications to other parameters? Providing a discussion on the connection of what is demonstrated or experiments expanding on this would add support for this claim.

- Claim: "Extensive experiments on 6 physical systems verify the effectiveness and efficiency of our approach."
  - While I do appreciate the experiments conducted on different systems. I feel like the use of "extensive" and "physical" are misleading with simulation-only approaches involving only two modest baselines. I would recommend reevaluating this claim or supporting the experiments with more baselines and hardware.

My understanding of operators in the control space is that they are global operators mapping cost functionals to another functional space. The authors do a great job of discussing that concept in section 3.1. However, I believe there is a disconnect when generating data and training the operator with a single initial condition. This process appears to produce an approximation to the operator that is only valid in a local region (starting from the initial condition or near). Does this process work from arbitrary initial conditions or is a separate trained network needed? My understanding is that it does not work from arbitrary initial conditions. If it does, I recommend adding a discussion supporting that and/or experiments that also show this attribute. If it does not work from arbitrary initial conditions, I am struggling to see the applicability to a control approach. For example, when using direct methods in an MPC fashion, we do solve for an open loop control sequence from an initial condition. However, we replan at each step. So while a lot of MPC-based methods assume deterministic dynamics when solving an open-loop problem, a lot of uncertainty is handled by the replanning at the new state at each time step.

Does the frequency of training data influence the performance at different control frequencies? For example, if we are wanting to control something at a rate much slower than its dynamics, would this approach still be valid to estimate the operator? Would the rate of data gathered for training influence this performance?


**Summary Of The Paper:**

This paper proposes a neural network architecture to estimate a mapping from cost functionals to optimal inputs in order to solve optimal control problems. The authors extend the work involving DeepONet (Lu et al., 2021 and Lanthaler et al., 2022) to optimal control problems and also extend the analysis to theoretically bound the error of their method, OptCtrlOP. The performance of OptCtrlOP was shown through simulation on six different systems ranging in states from one to nine dimensions and input dimensionality ranging from one to four. The provided results show an improvement in running time and accuracy against two baselines.

**Summary Of The Review:**

The concept of using a neural network as an operator mapping from cost functionals to control functionals is an important area of research. The paper is well written and the inclusion of both theoretical analysis and empirical results is a big strength. My primary concern with the paper as stands is the lack of support throughout for the very strong claims. Additionally, based on my current understanding of the approach, I do not see the novelty and how this approach would apply to control problems (if limited to a single initial condition). If this approach is the initial step towards work to provide a mechanism for a feedback/replanning process, then I recommend stating that more explicitly.

---

> ### Author Response · Authors · 2022-11-20
> **In response to Reviewer Yqn8**
>
> Thank you for the careful reading, insightful comments, and constructive suggestions. We appreciate that you acknowledged our work, including production quality, solid theoretical analysis, and rich empirical results. In the following response, we will provide detailed explanations to address your concerns. Probably due to system bugs in OpenReview, your review is not visible to us until Nov.17, thus our rebuttal time is very limited. If it is possible, we want to add more experiments in response to your questions and suggestions during the discussion phase.
>
>
> > ***Q1** Claim: "[OptCtrlOP] solves OCPs in a one-shot manner with no dependence on the explicit expression of dynamics or iterative optimization processes." But model training needs data, and data generation involves the expression of the dynamics or iterative nature.*
>
> **Re-1**: Maybe it is more strict to claim that "a trained OptCtrlOP solves OCPs in a one-shot manner without direct dependence on ...". We made this claim to show: 1)applicability (not necessarily depends on OCP expressions) and 2)inference efficiency (one-shot, not iterative).
>
> For the first point, OptCtrlOP not necessarily depends on OCP expressions, since the data can be generated by a  feedback (and model-free) controller. For example, in our newly added experiment on the Pushing dataset, the data is generated by an industrial robot arm pushing objects. The expression of dynamics is unavailable since the friction of the surface is not uniform.
>
> For the second point, it is common sense that data-driven methods necessarily involve data collection and training. Both steps may have iterative steps, and the efficiency is limited. But we do not emphasize the efficiency of those steps, since they can be performed offline, and their efficiency is not critical in the application (to our knowledge).
>
>
> > ***Q2**: Compare running time with SCP-based DM baselines. And how the inference time scales as the network size grows?*
>
> **Re-2**: As for SCP-DM, we try to implement DM by the built-in SCP (with qpoases backend) of casadi, but the running time dramatically drops to 1 sec./instance, along with some non-convergence warnings. We are struggling to find a more stable and user-friendly SCP package.
>
> And for the inference time scalability, a good example is the pushing dataset, where the input dimension to the network is 801 (including a 768-dim image), and the inference time of OptCtrlOP is $6 \times 10^{-6}$sec./instance, which is 10x of that of Quadrotor. And if the model size grows even larger, it is more reasonable to deploy the model on a GPU. Our model does not involve expensive computations like Fourier transforms or graph convolution, and we expect its running time scalability is similar to other general neural architectures.
>
>
> > ***Q3**: Need more baselines, and more experiments on the real-world environment.*
>
> **Re-3**: We add two neural operator baselines: FNO and GEN (see the update of section 3). And we perform an experiment on a real-world object-pushing dataset, which records the forces and motion of a controlled robotic arm pushing objects on various surfaces.
>
> > ***Q4**: Does OptCtrlOP generalize to different initial conditions and dynamics?*
>
> **Re-4**: We present two experiments to verify such generalization ability. The first one is the newly added experiments on pushing dataset, where the cost functional, dynamics and initial state are all variables. And the second one is the Stochastic Pendulum environment with a feedback control scheme, where OptCtrlOP learns the mapping from the current state to a single step of control. Due to the limitation of rebuttal time, we do not design more experiments. If you find it necessary to evaluate performance on other model parameters (e.g. length of the pendulum, weight, as you mentioned), we can add it during the discussion period.
>
> Update: we added an extended experiment on Quadrotor (see official comment), in which target states, dynamics, and initial states are all variable.
>
> > ***Q5**: Does the frequency of training data influence the performance at different control frequencies? For example, if we are wanting to control something at a rate much slower than its dynamics, would this approach still be valid to estimate the operator? Would the rate of data gathered for training influence this performance?*
>
> **Re-5**: We do not fully understand the problem setting here, especially the term "frequency of training data". Does it mean the density of time indices when discretizing the time horizon for DM (e.g. we now have 100 indices in the horizon T=[0,1])? And to our understanding, "control something at a rate slower than its dynamics" means that control is discretized with a lower density of time indices (e.g. 10 indices in the horizon). After the data is generated, the OptCtrlOP needs to learn such a piece-wise constant control function.
> Do we understand your points correctly? If possible, could you please provide more explanations?

---

> > ### Comment · Reviewer_Yqn8 · 2022-12-05
> > **Response to Author Comments/Revisions**
> >
> >
> > Thank you for the replies and the clarifications you have provided!
> >
> > **Q1** - I think this wording eases a lot of my concern. In your response, you state “OptCtrlOP not necessarily depends on OCP expressions”. I am a bit confused by this statement. From my understanding, the method presented doesn’t necessarily solve an OCP, but provides an approximate functional operator using demonstrations from other control methods (not necessarily optimal control as in the pushing example). So, if non-OCP samples are provided, I wouldn’t expect OptCtrlOP to achieve optimal control inputs. I may be misunderstand the approach, and if so, I apologize. From your statement, are you implying that OptCtrlOP can provide optimal control from arbitrary control demonstrations? The main point I was trying to bring up in the discussion was the reliance on some form of optimal control. If this isn’t true, that is a significant point and should be made more explicitly.
> >
> > **Q2** - I appreciate the look into existing SCP-based solutions. There is a lot of work being done with SCPs and I would recommend investigating different implementations for future work.
> >
> > **Q3** - My wording here might have been poor. I was trying to express concerns similar to other reviewers. Specifically the definition of reusability. I think this idea is critically important for control problems. While you demonstrate performance on small distribution shifts, where does this break down? Meaning when is the shift enough where the performance is unacceptable (I know “unacceptable is domain/problem specific, but I wanted to address the larger idea)? While networks are a great tool to approximate functions, they still experience failures when exposed to OOD data (not just a slight distribution shift). Since the idea of reusability is important and has been a discussion point of multiple reviewers, I recommend expanding on this a bit more.
> >
> > **Q4** - Thank you for providing the additional experiments! This is still my primary area of concern (related/connected to Q3). Can you expand a bit more on the differences in the initial conditions? In the quadrotor experiments provided on 24 Nov, there is approximately a 23 times increase in MAPE. I’m assuming you would expect performance to decrease as inputs are further shifted from the ID training data?
> >
> > **Q5** - I think you were interpreting my questions correctly. This isn’t a large area of concern, more of a question for discussion/clarity. I was referring to the discretization of the samples at a lower density (e.g. 10 indices in the horizon). If we have a system with a given frequency response, is there a sampling frequency required? This question came from thinking about the requirements of moving from classical control to digital control and the relation to time delay. I know these concepts are different but I was curious about your insights on the effectiveness of your method at different sample rates of the training data.
> >
> > Other - In the new manuscript, I think there might be a typo in 3.1.1 for the initial x conditions (section E.4 appears to have the appropriate dimensions).

---

> > > ### Author Response · Authors · 2022-12-10
> > > **In response to the reply from Reviewer Yqn8**
> > >
> > > Thank you again for your insightful questions. We will provide an extra experiment and more discussion in response to your concerns.
> > >
> > > > ***Q1**: From your statement “OptCtrlOP does not necessarily depend on OCP expressions”, are you implying that OptCtrlOP can provide optimal control from arbitrary control demonstrations?*
> > >
> > > **R1**: We are sorry for the confusing wording. If non-optimal samples are provided, OptCtrlOP can not output optimal control, since it is essentially a supervised model. It is true that "[OptCtrlOP] provides an approximate functional operator using demonstrations from other control methods", thus OptCtrlOP inevitably relies on some form of optimal control.
> > >
> > > At a high level, all optimal control algorithms and models should involve some form of optimal control, either explicitly or implicitly. Our statement “OptCtrlOP not necessarily depends on OCP expressions” only discusses the reliance on explicit mathematical OCP expressions.
> > >
> > > > ***Q2**: If the performance decreases as inputs are further shifted, then when is the shift enough where the performance is unacceptable?*
> > >
> > > **R2**:
> > > We test the OOD(out-of-distribution) MAPE on Quadrotor with more distribution shift on initial states, and display the results in the following table. The percentage numbers in the table header denote the magnification factors on the distribution shift. If we denote $\mathbf{x}^\text{out}_\text{init}:= \mathbf{x}_\text{init} + \epsilon_\text{out}$, then 100% corresponds to $\epsilon_\text{out}\sim  U(-0.1, 0.1)$, 200% denotes $\epsilon_\text{out}\sim  U(-0.2, 0.0)$,..., 500% denotes $\epsilon_\text{out}\sim  U(-0.5, -0.3)$. The ID(in-distribution) noise is fixed as $\epsilon_\text{in}\sim  U(0.1, 1.1)$.
> > >
> > > | Model |100%|200%|300%|400%|500%|
> > > |:------:|:------:|:------:|:------:|:------:|:------:|
> > > |OptCtrlOP |$2.01 \times 10^{-4}$|$5.23 \times 10^{-3}$|$2.13 \times 10^{-2}$|$6.36 \times 10^{-2}$|$1.95 \times 10^{-1}$|
> > > |MLP |$1.95 \times 10^{-4}$|$5.19 \times 10^{-3}$|$1.79 \times 10^{-2}$|$4.70 \times 10^{-2}$|$1.48 \times 10^{-1}$|
> > > |GEN |$5.05 \times 10^{-4}$|$1.14 \times 10^{-2}$|$1.37 \times 10^{-1}$|$5.58 \times 10^{-1}$|$1.51 \times 10^{0}$|
> > > |FNO |$1.24 \times 10^{-4}$|$1.76 \times 10^{-3}$|$1.19 \times 10^{-2}$|$3.54 \times 10^{-2}$|$8.89 \times 10^{-2}$|
> > > |PDP |$1.26 \times 10^{-4}$|$1.22 \times 10^{-4}$|$1.29 \times 10^{-4}$|$1.23 \times 10^{-4}$|$1.32 \times 10^{-4}$|
> > >
> > > From the table, one can observe that the performance of neural models decrease with more distribution shift, while non-neural method (PDP) performance remains the same. When the distribution shift increases to 500%, the results of neural models are unacceptable.
> > >
> > > The OptCtrlOP is mainly designed for ID reusability, although it can be reused on OOD with zero-shot (i.e. no training samples from OOD), under moderate distribution shift. For large distribution shifts, such zero-shot transfer learning is still a challenging task for neural models, since the information of OOD is almost completely missing.
> > >
> > > In practice, if large shift OOD reusability is required for some application, we may assume a few training samples from OOD is available (i.e. few-shot). And OptCtrlOP can be fine-tuned on the training samples from OOD to achieve better performance. This idea has been proposed for DeepONet few-shot transfer learning in [1], and we think it is an important future direction for OptCtrlOP.
> > >
> > > > ***Q3**: If we have a system with a given frequency response, is there a training data discretization frequency required?*
> > >
> > > **R3**: To our understanding, there are two types of frequencies involved here, which we term system frequency and discretization frequency respectively.
> > >
> > > Firstly, the system frequency is the maximal frequency of the exact solution of OCP, which is determined by the property of OCP itself, independent of solvers.
> > >
> > > Secondly, the discretization frequency is used by the reference OCP solvers, e.g. the DM solver in our synthetic experiments. Those solvers need to discretize infinite-dimensional OCP into an approximate finite-dimensional problem for computing. Intuitively, this frequency should be at least larger than twice the system frequency to capture all the information (Nyquist-Shannon sampling theorem). Meanwhile, this frequency also determines the error of the numerical OCP solver. If the discretization frequency is too small, then the solver may produce a highly biased solution. As discussed in Q1, OptCtrlOP, as well as almost all supervised neural models, can not learn accurate knowledge from poor data. It is an interesting future direction to provide the quantitative analysis of the relationship between discretization frequency and the error of OptCtrlOP. In the experiment of this paper, however, we simply fix the discretization frequency to 100Hz for all synthetic environments.
> > >
> > > **Reference**
> > > 1. Goswami, Somdatta, et al. "Deep transfer operator learning for partial differential equations under conditional shift."

---

### Official Review · Reviewer_vSM1 · 2022-11-05

**Confidence:** 3
**Correctness:** 3
**Technical Novelty And Significance:** 4
**Empirical Novelty And Significance:** 4
**Recommendation:** 6

**Clarity, Quality, Novelty And Reproducibility:**

Clarity: The clarity of the paper is low. I would rate this as a 2/10. Many specific reasons mentioned in the Strengths and Weaknesses list.

Quality: I'm not sure what this is intended to refer to. Overall, the technical work seems sound (as far as I can understand) but it is not polished.

Novelty: It is novel so far as I know.

Reproducibility: As I mentioned earlier, network architecture diagrams would be useful and help with reproducibility. As would improving the overall clarity of the paper.

**Strength And Weaknesses:**

The main strength of this paper is that the method is incredibly fast – the method is many orders of magnitude faster than more traditional methods. Speed is one of the main bottlenecks of optimal control approaches, so methods that obtain similar performance with much less computational cost are valuable.

The main weakness of this paper is that the writing presents the content hastily; with little exposition, explanation, or context, making it difficult to follow the logical argumentation. Despite being someone who does research in the area of control theory, I had to read several other papers just to understand this paper. The audience will never have exactly the same background and expertise as the authors and the paper should be written with that expectation. This is especially relevant because this is a controls paper submitted to a machine learning conference – the vast majority of registrants to ICLR have little to no background in control theory but it should be accessible to them, as they are the intended audience.

Specific points to address:
1. The second paragraph of the introduction jumps immediately into "Lebesgue measurable function space" with no definitions. I would ask – is this reference to the foundations of optimal control theory necessary? Or can it be omitted and instead a plain-language description of the optimal control problem added, after which the paper transitions directly to the formulation of the optimal control problem (Eq. 1)? This is the type of hasty writing without exposition, explanation, or introduction that appears throughout the paper and makes it difficult to read and understand.

2. The concept of model reusability is not defined but should be defined in the paragraph where it is mentioned on page 2.

3. The concept of two-phase optimal control solvers is not defined but should be defined in the paragraph where it is mentioned on page 2.

4. In the discussion of PINNs in section 2.1, the paper mentions that "the magnitude of the two loss terms is inherently imbalanced" but this has little meaning for the reader without the appropriate context. Which two loss terms? Why should the reader care about this imbalance? As a note for here and throughout the paper: the paper should summarize other works with enough context so that the reader can appreciate the points the paper is making without having to re-read every reference.

5.  It says in section 2.1 that for Direct methods of solving optimal control problems, "The
reformulation essentially constructs surrogate models, where the state and control function (infinite dimension) is replaced by polynomial or piece-wise constant functions." I do not understand what reformulation the paper is referring to. To my knowledge, using a direct method to solve an optimal control problem does not necessitate using a surrogate model. What am I missing here?

6. At the end of section 3.1 the paper claims that the approach is highly reusable, but as I mentioned above, there is no definition of reusability that the paper has given so it is difficult to evaluate this claim. On my first read through this paper, I understood this to mean that the cost function could change even if the dynamics had to remain the same. However, later in the methodology the paper defines the encoder as simply returning the target state and effectively learn with respect to a fixed cost function. It seems like a major simplification. Does this claim of reusability just refer to the fact that the goal point can change? (because it's an input to the network)? I think this point needs major clarification. If possible, it would be nice to see an experiment where the encode of the cost function is more sophisticated.

7. The main algorithm of the paper (Alg. 1) appears in the appendix. It would be nice to see it appear in the text, although I know space is limited. In contrast, the definition of a neural network in section 3.2 list item 2 (Approximator) is not necessary for the readers of ICLR papers who are all familiar with machine learning. It should be moved to the appendix.

8. I had difficulty following the theoretical proof in 3.3. Much more intuition, definition, and explanation could be added. The terms in eq. 6 are not well defined. What is a push forward measure? Even using Lip() to denote Lipschitz constant (I presume) should be defined.

9. For the pendulum example appearing in the text – I would encourage you to switch this out for something with more interesting dynamics that you have already run experiments on. It seems that every controls paper I read uses an inverted pendulum and it does not demonstrate the capability of the method. One would only use an involved learning based control algorithm like the one in this paper for a more complex system. Also, I believe there may be an error in the way in-distribution and out-of-distribution noise was defined for the pendulum – they appear to be switched, I think? It would make more sense for the in distribution noise to be centered around the goal point.

10.  As a high-level point of feedback – I believe that a more concrete description of the method is in order. E.g. perhaps a diagram of network architecture similar to the DeepONet paper as well as a straightforward description like this line from the DeepONet paper: "Let G be an operator taking an input function u, with G(u) being the corresponding output function. For any point y in the domain of G(u), the output G(u)(y) is a real number. Hence, the network takes inputs composed of two parts: u and y, and outputs G(u)(y) (Fig. 1a)."

11. Because the scale of the plots is logarithmic, the reader cannot easily verify a claim like the method is "50% faster" by e.g. looking for a bar that is 50% larger than another.

12. The paper discusses the performance gap between ID and OOD performance but does not give numbers and also plots these on separate graphs, so again, it is hard for the reader to validate these claims.

13. The paper gives a *hypothesis* as to why your architecture is better at handling OOD shift, but it is stated more that you *know.* I would clarify that this is an explanatory hypothesis.

14. In the abstract, six experiments are mentioned, but the plots in Fig. 2 omit the heat problem. Why?

15. It is difficult to extract information from fig. 4. Perhaps you could use 2D plot views instead? Thinner lines? Euclidean distance from the optimal path?

16. The paper does not mentioned limitations of the work in the conclusion, but should.


**Summary Of The Paper:**

In this paper, the authors propose a new, data-driven method to solve optimal control problems that is significantly faster than existing methods at execution time and produces similar suboptimality of solution when compared to other approaches. The method is explained through the lens of learning an operator that maps an optimal control problem to a control function. Specifically, the authors learn a neural network that maps the goal state and the time index to a control input for the dynamical system. The network architecture is inspired by the DeepONet paper.

**Summary Of The Review:**

To summarize, I think that this is exciting work but the current presentation does not do it justice. Were this an option I would give the recommendation of: accept conditioned on major revision.

---

> ### Author Response · Authors · 2022-11-20
> **In response to Reviewer vSM1**
>
> Thank you for the careful reading, nice comments, and constructive suggestions. We are grateful that you appreciated our work, including the efficiency, novelty, and theoretic soundness. In the following response, we will provide detailed explanations to address each of your concerns point-by-point.
>
>
> > ***Q1**: Definition of Lebesgue function space, reusability, and two-phase control.*
>
> **Re-1**: Lebesgue space is the normed function space, a generalization of the concept of normed vector space. We introduce the property of Lebesgue space in the analysis of the approximation error. As pointed out by the reviewer, it can be omitted in the introduction part. The reusability is measured by the capability of utilizing historical data when facing an unprecedented problem instance. And two-phase control consists of learning a neural surrogate model (phase-1), and solving the optimization problem utilizing the surrogate model (phase-2). We have clarified those two definitions in the main text.
>
> > ***Q2**: More explanation of 1) two loss terms of PINN and 2) reformulation of direct methods.*
>
> **Re-2**: The two loss terms of PINN are the residual loss and boundary condition loss, and the consequence of the imbalance of these terms is that PINN may not converge to the correct solution. Take our OCP setting, for example, we have tried to solve the OCP by a vanilla PINN, but it fails to output correct prediction even for a simple pendulum problem.
>
> The reformulation of direct methods is to convert the original OCP (infinite dimension) into non-linear programming (finite dimension), e.g. by discretizing the dynamics and cost functional. And such discretization (or other reformulation methods) constructs a surrogate of the original system.
>
> > ***Q3**: Does the OptCtrlOP only generalize on different target trajectories, and only for simple dynamics? Can OptCtrlOP incorporate a more sophisticated encoder?*
>
> **Re-3**: We switch the Pendulum experiment for the Quadrotor, whose dynamic functions are more complicated. In addition, we perform an experiment on a real-world object-pushing dataset, to evaluate the generalization ability of OptCtrlOP. The dataset records the forces and motion of a controlled robotic arm pushing objects on various surfaces. In our experiment, the cost functional, dynamics, and initial state are all variable, and the ID/OOD is defined on the initial state. The dynamics of the pushing problem are hard to model since the friction of the surface is not uniform. In such a complex task without explicit expression, the classical direct or indirect methods, such as DM and PMP, are not applicable. However, OptCtrlOP is able to learn from demonstration data, providing relatively accurate predictions compared with all neural baselines. In this problem, the encoder consists of various components, such as CNN (to encode the images of object shape), smoothing filter (to denoise the trajectories), etc.
>
>
> > ***Q4**: More definitions and explanations for derivation of theorems, e.g. definition of push-forward measure.*
>
> **Re-4**: We have added some intuitive explanations in section 2 for better readability. Informally, the push-forward measure is a measure obtained by transferring (i.e. "pushing") another measure by a function. We add the formal definition of push-forward measure, as well as the Lipschitz constant, in the main text.
>
> > ***Q5**: A more concrete description of OptCtrlOP is in order, e.g. a diagram, a straightforward description, or an algorithm box.*
> >
> **Re-5**: We update the diagram and description (Fig. 2). The algorithm box is left to appendix since its information is covered by the diagram.
>
> > ***Q6**: The definition of ID/OOD noise appears to be switched. It would make more sense for the ID noise to be centered around the goal point.*
>
> **Re-6**: In our definition of ID/OOD, the concept of in and out is characterized by the distance with the distribution of training data, not the distance with the baseline target state $\mathbf{x}^{goal}_{base}$. In other words, a test set coming from a different distribution with training data is called an OOD test set, while a test set sampled from the same distribution as training data is called an ID test set.
>
> > ***Q7**: The performance plots are logarithmic, hard to compare results quantitatively. And ID-OOD gap is not reported explicitly in numbers.*
>
> **Re-7**: We add tables listing the numerical results (Tab.2 and Tab.6-10 in the Appendix).
>
> > ***Q8**: The reason for the OOD generalization ability of OptCtrlOP is only a hypothesis, not a fact. And limitation part is missing in the conclusion section.*
>
> **Re-8**: We modify the statement on the reason for OOD generalization, clarifying that it is an unproved conjecture. And in the conclusion section, we add a paragraph discussing the limitation, including the analysis of OOD generalization, and problem-specific network architecture design.

---

> ### Author Response · Authors · 2022-11-20
> **In response to Reviewer vSM1 (continued)**
>
> > ***Q9**: Why Heating problem is missing in the plot?*
>
> **Re-9**: The heating problem is a PDE-constrained OCP, while the 5 problems shown in the plot are all ODE-constrained OCPs. Considering the page width, we only visualize those 5 ODE-OCPs, leaving the heating problem to Tab.10 in the appendix.
>
> > ***Q10**: It is difficult to extract information from the trajectory plot (Fig. 4 of original submission)*
>
> **Re-10**: We leave the trajectory plots to Fig.9 in the appendix. We decide to treat those plots as merely a demonstration, instead of a performance comparison, since the number of samples is very limited.

---

### Author Response · Authors · 2022-11-19
**Paper revision summary**

We thank all reviewers for the insightful feedback and suggestions for improving the quality of the paper. Below is a summary of the major revision (also marked as blue in the main text):
1. [Section 1] Clarify definitions of model reusability and two-phase models and remove the Lebesgue space, as suggested by reviewer vSM1.
2. [Section 1] Move related works to Appx. B. Add FNO to related works of the neural operator, as required by reviewer p9mD.
3. [Section 2] Add a model diagram (Fig. 2), along with a straightforward description. Add more explanation and intuition for theorems and definitions, as suggested by reviewer vSM1.
4. [Section 3] Add two neural operator baselines (FNO and GEN), and compare their accuracy and running time with our model (Fig. 3),  following reviewer p9mD’s suggestions.
5. [Section 3] Switch the details of the Pendulum experiment for the Quadrotor experiment (Fig.4 and Tab.2), as advised by reviewer vSM1 and p9mD.
6. [Section 3] Move the figure of trajectories into Appendix (Fig. 9).
7. [Section 3] Present running time, accuracy, and ID\OOD gap numerically in tables (Tab.2 and Tab.6-10 in Appendix), as suggested by reviewer vSM1.
8. [Section 3] Add an experiment on a real-world dataset, where the control of a robot arm pushing objects is learned (Fig.5 and Tab.3). The task involves variables including target state functions, dynamics, as well as initial states, following all reviewers’ suggestions.
9. [Section 4] Add a future work (limitations) paragraph, as required by reviewer vSM1.

---

### Author Response · Authors · 2022-11-25
**Extended experiment on Quadrotor environment**

Extended experiment on synthetic environment Quadrotor, with variable target states, initial conditions, and dynamics, as suggested by Reviewer Yqn8 and Reviewer vSM1.


| Model | Time(sec./instance) | ID MAPE  | OOD MAPE |
|:------:|:------:|:------:|:------:|
|DM | $1.59 \times 10^{-1}$ | $\diagdown$  | $\diagdown$  |
|OptCtrlOP | $8.39 \times 10^{-7}$ | $3.55 \times 10^{-5}$  | $8.15 \times 10^{-4}$  |
|MLP | $8.45 \times 10^{-5}$ | $4.71 \times 10^{-5}$  | $8.36 \times 10^{-4}$  |
|GEN | $1.10 \times 10^{-4}$ | $2.20 \times 10^{-4}$  | $5.09 \times 10^{-4}$  |
|FNO | $1.95 \times 10^{-3}$ | $6.00 \times 10^{-5}$  | $1.26 \times 10^{-4}$  |
|PDP | $7.25 \times 10^{1}$ | $1.13 \times 10^{-4}$  | $1.15 \times 10^{-4}$  |

The results verify that OptCtrlOP has generalization ability not only for target states, but also for initial conditions and dynamics parameters.

---

> ### Author Response · Authors · 2022-11-29
> **Looking forward to further discussion.**
>
> Thank you again for the thoughtful reviews. Do our responses and newly added experiments address your concerns? We are looking forward to your valuable comments. Please let us know if you have any follow-up questions.

---

### Author Response · Authors · 2022-12-11
**A kind reminder of discussion**

Dear Area Chairs and Reviewers,

We would like to sincerely thank you again for your time, efforts, and your insightful reviews!

We hope that our responses and updated draft and experiments have addressed all major concerns. It would be appreciated if we had the chance to respond to your further questions. As the deadline for the final discussion stage (Nov.12) is approaching, would you mind starting the discussion for our paper (if not done yet)?

Best regards,

Authors.

---

### Decision · Program_Chairs · 2023-01-20

**Decision:**

Reject

**Justification For Why Not Higher Score:**

The paper was borderline and there was discussion around it well into the review period. However, the reviewers still had concerns around the quality of writing that prevented the paper from being accepted in it's current form.

**Justification For Why Not Lower Score:**

N/A

**Metareview: Summary, Strengths And Weaknesses:**

The authors develop a new, data-driven method to solve optimal control problems that is more efficient than existing methods at execution time and achieves similar performance.

Strengths:
1. Substantial speedup based on learning an operator that maps an optimal control problem to a control policy.
2. Thorough experimental comparisons to several baselines (particularly after the rebuttal phase).

Weaknesses:
1. Clarity of writing makes the contributions of the paper hard to read and appreciate.
2. The performance degrades under substantial distribution shift.

Overall, the paper has some interesting contributions, but the weaknesses make it borderline overall. I think the paper would benefit from a thorough revision and resubmission to a future venue, taking the feedback from the reviewers into account.

**Summary Of Ac-Reviewer Meeting:**

No meeting